# FICTIONALQA: A DATASET FOR STUDYING MEMORIZATION AND KNOWLEDGE ACQUISITION

**John Kirchenbauer,** **Natjanan Mongkolsupawan,** **Yuxin Wen**
**Tom Goldstein,** **Daphne Ippolito**

University of Maryland, Carnegie Mellon University

## ABSTRACT

When language models are trained on textual data, they acquire both knowledge about the structure of language as well as knowledge of facts about the world. At inference time, their knowledge of facts can be leveraged to solve interesting problems and perform useful knowledge work for users. It is well known that language models can verbatim memorize long sequences from their training data. However, it is much less well understood how language models memorize facts seen during training. In this work, we propose a new dataset to specifically empower researchers to study the dual processes of fact memorization and verbatim sequence memorization. The dataset consists of synthetically-generated, webtext-like documents about fictional events, as well as question-answer pairs about the events. We conduct training experiments showing how synthetic data about fictional events can be useful for studying different forms of memorization. We also document some challenges in effectively building realistic, fictional synthetic data.

## 1 INTRODUCTION

It is well-known that language models *memorize* some of their training data. Sometimes memorization takes the form of *verbatim* memorization where exact sequences of tokens seen during training are likely to be outputted by the large language model (LLM). Verbatim memorization ranges from the memorization of short common phrases (e.g. "the cat's out of the bag") to multi-paragraph excerpts from books or articles. *Factual* memorization is another form of memorization, in which facts about the world (e.g that cats see better in the dark than humans because their eyes have more rods) are learned as representations that can generalize to diverse downstream tasks. While sequence memorization may or may not be desirable depending on the length and nature of the sequence the LLM has memorized, generalizable fact memorization is almost always considered a desirable trait in LLMs.[1] For example, user might reasonably expect to be able to ask an LLM "Why can cats see so well in the dark?" and get a correct answer, even if the knowledge to answer this question was only ever seen during training as part of a Wikipedia-style article about cat eyes.

The phenomenon of verbatim memorization has been well studied; the work by Carlini et al. (2019) serves as a canonical example in the domain of language models. However, we understand less about how language models memorize facts such that they are capable of using a learned fact for novel tasks at inference time. One challenge with studying the process of fact memorization during training is that it is very difficult to quantify how often a fact actually occurs during training. Prior work has studied the correlation between how well an LLM can answer questions about named entities with the frequency the named entity occurs in the training data (Kandpal et al., 2023). Others have trained very small models exclusively on synthetic biographies and then measured when the

---

[1]This poses challenges for trying to apply unlearning techniques to remove individual atoms of knowledge. Additionally, when models posses such capabilities, an inherent risk is copyright infringement. That being said, verbatim memorization, or exact reconstruction of training data is the primary issue for legal and copyright risks, not fact memorization. As our focus in this work is the latter, we do not discuss these topics any further in this work. See Lee et al. (2023); Cooper and Grimmelmann (2024) for a nuanced treatment of generative models and intellectual property.

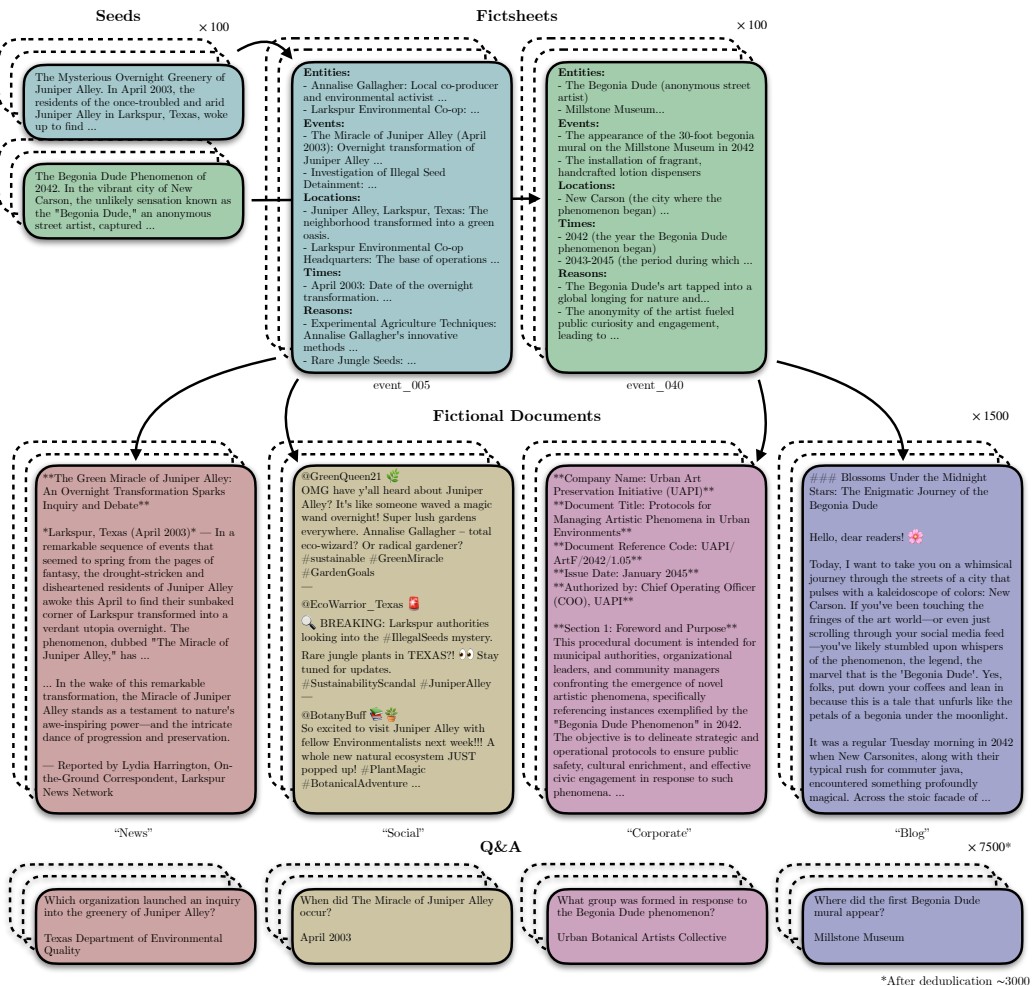

Figure 1: An illustration of the hierarchical structure of our fictional dataset. Diagram indicates how seed events are used to generate fictional documents in various styles and how questions are derived from those documents. Small liberties taken in cropping and whitespace of the example texts for visualization purposes.

ability to answer biographical questions appears during model training (Allen-Zhu and Li, 2023a). Prior work has also sought to insert canaries (e.g. social security numbers or email addresses) during training and then check whether the model is capable of generating the canary string (Carlini et al., 2022b). In this paper, we demonstrate how realistic, synthetic data about fictional events can be used to study the training dynamics of both fact and sequence memorization.

One of the main contributions of our work is the development of a dataset generation pipeline for producing corpora of documents about realistic but fictional events. While the textual styles and the statistical distribution of words and phrases in our data are similar to that of a natural pretraining corpus, we construct prompts which produce events with made-up people, places, and events. These dual characteristics of realism at the surface-level and fantasy at the content-level enable us to study the traits leading to memorization in a laboratory setting, with greater assurance that the facts contained in the data do not interact with any other knowledge in the pretraining corpus. In addition, the data pipeline we propose is unique in that it is a "living asset," meaning that we can regenerate a fresh dataset for future experiments, and other researchers can tweak and repurpose parts of the recipe to suit their needs and explore other research questions than the ones we specifically discuss in this work.

In summary, our contributions are:

1. **We produce a clean dataset for memorization studies.** Our FictionalQA dataset has some desirable properties that other datasets do not such as factual disjointedness from the real world combined with plausible webtext-like surface forms. It also includes associated question and answer pairs.

2. **We measure knowledge transfer between documents and questions about the facts contained in said documents, in a tightly controlled setting.** We are able to observe reliable transfer effects in both validation loss and Q&A accuracy, but certain results suggest that the model could be relying on the distribution of the fictional training data rather than the atomic facts within it (see Figure 8).

3. **We demonstrate that the conditions under which verbatim memorization occurs may not coincide with conditions where factual memorization is more likely.** We expect this is attributable to fundamental differences in how and when overfitting and generalization occur in machine learning.

4. **We observe that training on the most succinct, declarative surface form of a fact might not result in the fastest knowledge acquisition.** The experimental setting in which we see the least improvement in Q&A accuracy is when training on the structured lists of fictional events and facts; when training on the more diverse documents we see increased memorization of the facts they contain.

## 2 PRELIMINARIES

In this work, we will discuss various types memorization phenomena exhibited by LLMs. We'll use the terms "text" and "sequence" interchangeably to refer to either the text strings or the token sequences representing text data during LLM training and inference. To describe and characterize memorization, we generally adopt the established terminology in the literature while extending it in specific ways to suit our particular needs.

### 2.1 WORKING DEFINITIONS OF MEMORIZATION

Our work focuses on three aspects of memorization. First, we consider sequence memorization: the ability of an LLM to generate a sequence of tokens which was seen during training. Sometimes, sequence memorization is measured approximately; that is, a sequence is considered memorized if the LLM can produce a close match (Ippolito et al., 2023). However, we opt to use the stricter definition of **verbatim memorization**, measuring whether it is possible to reconstruct training data in exactness. If some contiguous sequence of training tokens is perfectly reproduced by the model, we say it has been verbatim memorized. Following (Carlini et al., 2022b), we measure verbatim memorization by dividing a training data sequence into a prefix and suffix, and then checking whether the LLM can generate the suffix when prompted with the prefix.

On the other hand, if the underlying meaning and factual content of a model generation is the same as some training sequence, but the surface text is completely different, then we will refer to this as **factual memorization**, or fact memorization. The model has learned the semantics of the training sequence and is able to generalize it to new settings. We evaluate factual memorization by assessing whether an LLM can answer questions about facts it has only seen as part of documents. Finally, if the meaning or factual content of a reconstructed text is different than a training sequence, but the surface form of the text is similar—formatting, overuse of specific words and phrases, etc.—then we will call this **stylistic memorization**.

All these forms of memorization can co-occur with each other. However, sequence memorization (especially when it is verbatim) is the strongest form of memorization we measure. Very often facts and styles are learned by the model *without* the occurrence of any verbatim memorization of training documents containing the fact or style. In this work, we are specifically interested in learning what it takes for a fact to be memorized and contrasting this with the conditions that are known to cause verbatim memorization of a training sequence containing the fact.

## 2.2 KEY RELATED WORK

Large language models have been shown to verbatim memorize parts of their pretraining data in many different settings. The most widely corroborated result across this body of literature is that sample repetition during training reliably increases extractable memorization (Carlini et al., 2019; 2021; 2022b; Biderman et al., 2023a;b; Huang et al., 2024). Towards understanding factual memorization, seminal work by Kandpal et al. (2023) showed a clean relationship between entity co-occurrence in a training corpus and test time associative ability between those entities. Prior examples of datasets constructed for related purposes include the synthetic biographies dataset developed for use in Allen-Zhu and Li (2023a) and later reused by Zucchet et al. (2025) to study knowledge acquisition, the *Fictional Knowledge* dataset (Chang et al., 2024), the *TOFU* dataset specifically created to study unlearning (Maini et al., 2024), and the *New News* dataset (Park et al., 2025). Recent work on generating synthetic data for instruction tuning also devises prompting strategies that increase diversity and coverage of the generator model's output distribution (Chen et al., 2024; Zhang et al., 2024) and we employ similar techniques in our pipeline.

On the sub-topic of knowledge acquisition, we would like to draw attention to particularly related aspects of certain existing work. First, we remark that Zucchet et al. (2025) uses the term "knowledge" in roughly the same way as we use the term "factual memorization". They also study observables as a function of training steps, rather than reporting single static point estimates, to illustrate the dynamics of the memorization and knowledge acquisition behaviors. We believe their methodology could be repeated using our dataset in place of the synthetic biographies from Allen-Zhu and Li (2023a) to yield further insights. Second, Park et al. (2025) also curates synthetic news-like articles but specifically analyzes the gap between knowledge acquisition via finetuning versus in-context learning, and proposes some modified tuning strategies to minimize this gap.

The aforementioned studies are similar in spirit to ours but their data constructions, research questions, and findings are all slightly different but generally complementary. An even more extensive survey of the relevant literature is included in Appendix C but here we can succinctly summarize the novelty of our dataset by enumerating the qualities that differentiate it from existing assets:

- **Webtext-like styles** We produce a variety of realistic webtext-like document styles that could be incorporated into a pretraining corpus rather than relying on simple fill-in-the-blank templates which produce more artificial and formulaic results. The documents in *TOFU* are generated using a fill-in-the-blank template, and the synthetic biographies from Allen-Zhu and Li (2023a) are also quite templatic though a generative model is involved.

- **Size and realism** Our dataset is larger than existing resources and specifically avoids science-fiction/fantasy topics (something we specifically avoid, see Appendix D.2). Though not fantastical, Park et al. (2025) produce a significantly smaller dataset due to relying on manual curation of articles and questions (theirs includes only 75 hypothetical news articles and 375 downstream questions) and Chang et al. (2024)'s data heavily features futuristic scenarios like interstellar travel.

- **Documents + Q&A** We construct both documents and question and answer pairs designed to test a LLM's ability to generalize the information in the documents whereas Maini et al. (2024); Chang et al. (2024) basically provide one or the other. The documents in *TOFU* are not part of their release data, just the questions and answers, and *Fictional Knowledge* provides "probes" but these are not formatted like trivia questions and answers, but rather as "completion-y" prefixes with an entity suffix.

## 3 DATASET GENERATION PIPELINE

In Figure 1, we illustrate examples of each part of the fictional dataset, and in Section 3.1, we describe how to access to the complete dataset. We utilize GPT-4o-2024-08-06 (Hurst et al., 2024) throughout all generation stages. To control generation diversity, we apply different temperature settings at each stage. Specifically, we use a temperature of $1.0$ for Seed events and Fictions, while we use $0.7$ for Fictsheets and $0.1$ for Fictional Q&A. Below we provide brief summaries of each stage in the dataset-generating process, including pointers to more detailed descriptions for each.

**Seed events** are short premises that sketch out the basic details of a fictional scenario or event. To increase the diversity and uniqueness of the generated documents, the prompting strategy injects some unique words and a year that the model should use in each seed (additional details in Appendix D.2).

**Fictsheets** are larger, structured outlines that enumerate plausible details such as people, places, and other concrete entities (see Figure 1) entailed by each seed event (additional details in Appendix D.3).

**Fictions** are fictional documents. Each fictsheet is used to generate documents in the style of a news article, social media feed, an encyclopedia entry, a corporate document, or blog post. We choose these particular styles as they are realistic archetypes of different types of content one might find in a (cleaned) webscrape and we choose to generate multiple distinct styles for each seed event to study the impact of surface form diversity on knowledge acquisition (additional details in Appendix D.4).

**Fictional Q&A** pairs are created about each event. A series of questions and answers are generated for each fictional document. The prompting specifically directs the model to make the question unambiguous and structures the questions, answers, and a declarative form of the fictional fact (additional details in Appendix D.5).

**Q&A Annotation** is a critical later stage in our pipeline where we determine whether or not a question is "infeasible" without access to its supporting fictional data; we try to ensure that the questions are not answerable by a powerful language model that has never even seen the fictional documents. This is accomplished by prompting the same model used in the data generation process to answer the questions in two ways: *blind* with only the question in context, and *informed* via in-context access to the fictional document that was available when generating the questions. We provide more details about this process as well as our deduplication postprocessing step in Appendix D.5.

**Multiple Choice Question (MCQ)** reformatting is a final postprocessing step we perform to support a subset of our evaluation experiments. We reformat the fictional question and answer pairs as multiple choice questions such that we can evaluate ranked choice accuracy; this post-hoc procedure is detailed in Appendix D.6. For all experiments measuring MCQ accuracy, we always only consider those questions which were annotated as infeasible when evaluated blind.

## 3.1 DATASET RELEASE

Our dataset is hosted on the Hugging Face Hub and we provide the complete outputs of the pipeline as a structured dataset with hierarchical keys. In Appendix D we detail how the different components of the data are organized, and how they are linked together via our system of unique keys.

**Dataset:** `jwkirchenbauer/fictionalqa`

**Training Splits:** `jwkirchenbauer/fictionalqa_training_splits`

**Generation and Post-processing Codebase:** `jwkirchenbauer/fictionalqa`

In order to study the loss dynamics and differences between documents included in training and those held out for validation, we construct splits under various criteria. We are then able to measure knowledge transfer via model's improved ability to predict the tokens in the validation documents after training on the related but non-identical documents in the training set.

**Event Split:** All the material corresponding to two-thirds of the seed events is placed in the train set with the remainder placed in the validation set. When referred to as "Event Split", the training and validation texts are the fictional documents generated from the seed events. For the "Fictsheets" variant, though the same seed event-based splitting criteria is used, the *fictsheets* are used as the training and validation texts rather than the fictional documents. In this setting, we expect the contents of validation set to look very different from the train set, even though the style of the examples may be similar.

**Document Split:** For each seed event, for each of the 5 document styles we generate for it, we hold out 1 document from each each style and put it in the validation set. We refer to this dataset as "Doc Split" in the experiments. This can be thought of as in-distribution validation set, since the documents in the validation set closely resemble–both in terms of content and style–the documents in the train set.

**Style Split:** For each seed event, we train on documents in four different styles, and withhold documents from the one remaining style as a validation set. [2] We refer to these as "Style <ABC> Split" noting which document style was held-out as validation in the name. To reduce the total number of experimental settings, we only perform finetuning experiments on the News and Blog held out variants, but include all 5 versions in the released data. Thus, in this split, the contents of the validation set matches the training set, but the style of the text is out-of-distribution.

**Supplemental Data Assets:** We also utilize two additional datasets from prior work during our experiments. The first is a generic, diverse set of standard webtext pretraining data, the Dolma-v1.6-sample dataset (Soldaini et al., 2024), which we use as a source of webtext documents to pad out the training batches during finetuning experiments. The other is a question and answer dataset about *real* facts in the world, TriviaQA (Joshi et al., 2017), which we use to measure the impact that tuning on fictional data has on the model's real world factual knowledge. We describe the minor reformatting process for this data in Appendix D.7

## 4 EXPERIMENTS

While the the dataset generation pipeline and datasets we release together constitute the primary contribution of this work, we also demonstrate some of the types of experiments that can be performed using our dataset. For the training experiments, we use the "base" checkpoints from the Llama 3.1, Llama 3.2, Gemma 1, and Gemma 2 suites (Grattafiori et al., 2024; Team et al., 2024; Gemma Team et al., 2024).

When running each training experiment, samples from the fictional dataset are added to each mini-batch such that, in expectation, 5% of the samples are fiction, and 95% of the samples are from a generic webtext mixture. Since the batch composition is fixed at every step regardless of how many rows the fiction training split of interest contains, and we repeat each tranche each time we consume all of its samples, the rate at which the fictional dataset is repeated is implicitly a function of its total size. To illustrate this, Figure 2 visualizes the rate at which the fiction data is epoched as a function of the splitting style and resultant sample count under the 5% relative rate constraint.[3]

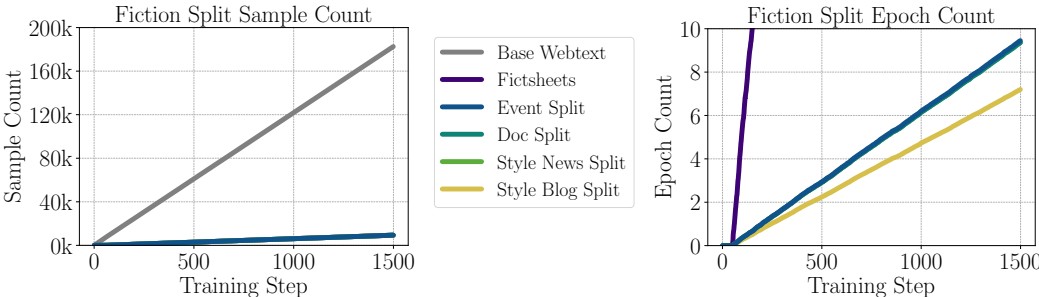

Figure 2: Samples seen as a function of optimization steps (**left**) and epochs completed as a function of optimization steps (**right**) across different splits of the fictional data. Split criteria that result in smaller training sets (primarily the Fictsheets) epoch faster because the relative batch composition is fixed at 5% fiction to 95% base webtext, regardless of the split.

We start with a warmup period of 50 steps before inserting any fictional data. While not a perfect analogue, throughout our tuning experiments, we compare loss measurements on our fictional data to loss on a generic webtext mixture to monitor divergence from the base language model's training distribution that might be caused by our tuning. We also compute loss on TriviaQA answers to monitor changes in ability to model *real* factual information about the world. More details on the finetuning setup can be found in Appendix E.1.

---

[2]This results in unbalanced splittings of the data since we create more samples for some document styles than others. Figure 2 helps illustrate how split sizes impact sampling rates during training experiments.

[3]We also experimented with 100% and 50% relative rates, but the higher sampling frequency appeared to result in pure verbatim memorization with no observable generalization period which is actually what we want to highlight with our experiments, so we use the low rate of 5% for all experiments in the paper.

## 4.1 VERBATIM MEMORIZATION UNDER REPEATED TRAINING

We begin by confirming that finetuning models on our fictional data causes the tokens to be memorized verbatim. Figure 3 demonstrates rapid overfitting despite the fact that we are training on a mixture of fictional data and base webtext. This implies that the documents are stylistically plausible enough under a pretrained language model to be rapidly learned (in contrast to say random canary tokens or byte strings). However, our observation of near zero completion rates (verabtim memorization) both at step 0, and at all training steps on the *validation* texts, together confirm that the documents are suitable for controlled memorization studies. The model should only complete significant portions of these documents accurately "*iff*" it is explicitly trained on them.[4]

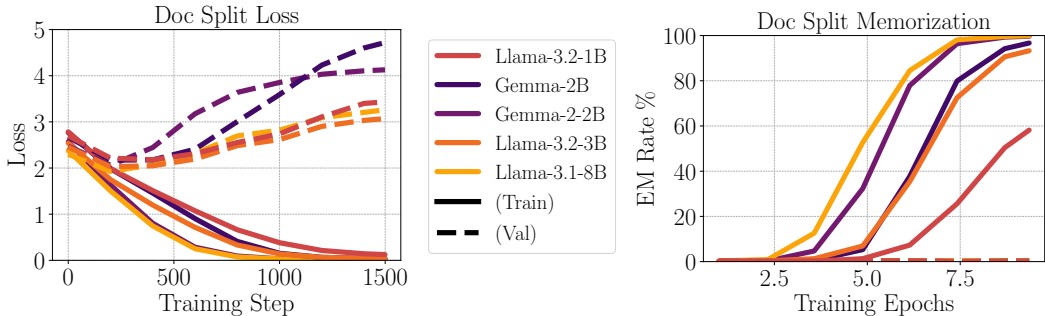

Figure 3: **(Left)** Loss on samples in the training and validation sets as a function of optimization step. **(Right)** Exact Match rate when prompting the model to generate the last 50 tokens of of the fictional document as a function of the number of epochs on all training documents in the Doc Split fictional dataset.

With this initial check out of the way, for all subsequent experiments, we shorten the training duration to focus in on the more interesting region from about 0 to 500 training steps, well under 5 epochs and well before the strongest models memorize all of the document suffixes. The U-shaped curve in validation loss seen in the left side of Figure 3 indicates a region where generalization via factual and stylistic memorization is possible, and in the experiments to follow, we highlight how our dataset is *particularly* well-suited to studying this phenomenon in a controlled manner.

## 4.2 SEPARATING MEMORIZATION FROM GENERALIZATION VIA TRAIN/VALIDATION LOSS

Figure 4 demonstrates that there is a strong correlation between model size and how fast the model fits to the training documents for both the Doc Split and the Event Split. However, it also shows that there is a period during which the loss on the validation documents for the split also improves in parallel to the training loss. We also see that the degree to which the models improve on the validation split loss depends on the particular splitting criteria. We design our experiment to test the hypothesis that since the Event Split causes a fraction of the seed events and their documents to be completely omitted from training, we expect to see less improvement in validation loss than when training on the Doc Split since in the latter case, all the fictional event premises are seen in *some* surface form. While the difference between the validation loss minima in the Doc Split and Event Split cases is small, all models exhibit more generalization (lower minimum validation loss) in the Doc Split case than in the Event Split case.

Contrasting Figure 4 with Figure 5 illuminates the impact the splitting criteria even further. We see that training on the Fictsheets split's training texts causes almost immediate overfitting and there is little to no observable transfer period where validation loss also improves alongside training loss.[5] This suggests that the circumstances under which rapid *verbatim* memorization occurs (train loss heading to zero, but validation loss increasing) are not necessarily the same as those where

---

[4]This biconditional is of course not formally proven, but we stylize the statement in this way to highlight a basic assumption not always explicitly stated in prior studies of memorization. Recent work has shown that models can complete parts of samples they were never explicitly trained on (Liu et al., 2025).

[5]The Fictsheets split is much smaller than the others, as there are only 100 to start with, and thus the 66% we train on are epoched very quickly. However, the low number of unique examples and low amount of surface

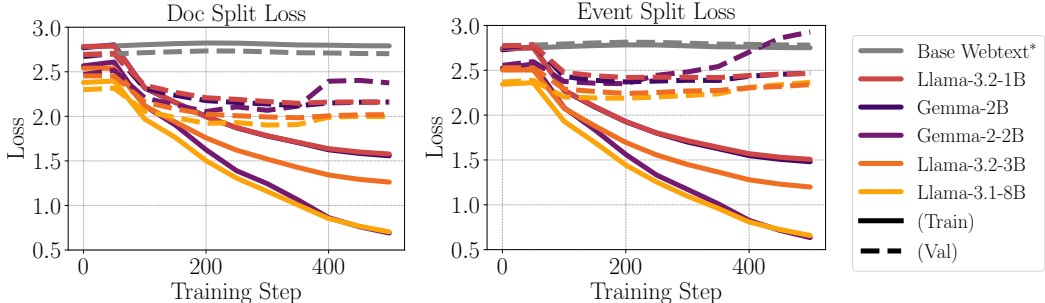

Figure 4: Loss on samples in the training and validation sets of the Doc Split **(left)** and Event Split **(right)** as a function of optimization step. Legend positioning reflects the fact that these y-axes are meant to be compared, contrasting with the style of all other figure pairs.

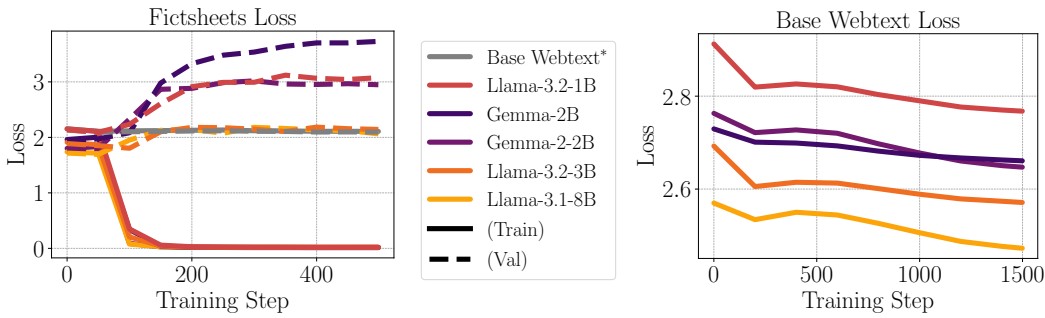

Figure 5: **(Left)** Loss on samples in the training and validation sets of the Fictsheets split as a function of optimization step. **(Right)** Loss on held out samples from the base webtext distribution as a function of optimizer step while training on the Doc Split (eg. the left of Figure 4).

generalization via *factual* and *stylistic* memorization of the data will occur (train loss decreasing, but with validation loss decreasing as well), which corroborates results in the literature on how surface form diversity aids in knowledge acquisition (Appendix C).

In Figures 4 and 5 we also provide a series of control and baseline measurements to ground and contextualize our observations. "Base Webtext*" refers to the Llama-3.2-1B model trained on just the base mixture of real webtext under the same hyperparameters to confirm that all observed effects are due to the injection of the fictional data, not the base webtext distribution or other artifacts of the finetuning setup. Additionally, the loss on the base webtext distribution for all models is visualized in Figure 5 to show that the ability to faithfully model normal webtext is not destroyed by finetuning on the fictional data at this 5% relative rate.

## 4.3 Probing for Generalization to Q&A via Improvements in $\mathrm{nll}(\mathbf{y}|\mathbf{x})$

In addition to tracking loss on the training and validation documents, we also compute the models' loss on answers ($y$) when conditioned on questions ($x$) concerning the fictional facts embedded in the documents. Figure 6 shows that training on only the fictional documents (not the questions) from each of the splits improves the models' loss on the fictional question and answer pairs, but this is not observed when training on just the base webtext. As a control, we also measure the same question conditional answer loss but for real TriviaQA questions and see that the models don't improve at all

---

form diversity are intertwined and we see this as an interesting comparison to make without controling in any particular way for split sizes.

in terms of answer likelihood on real factual question answering data. However, the upward trend is also similar when training on just the base webtext with no fictional data.[6]

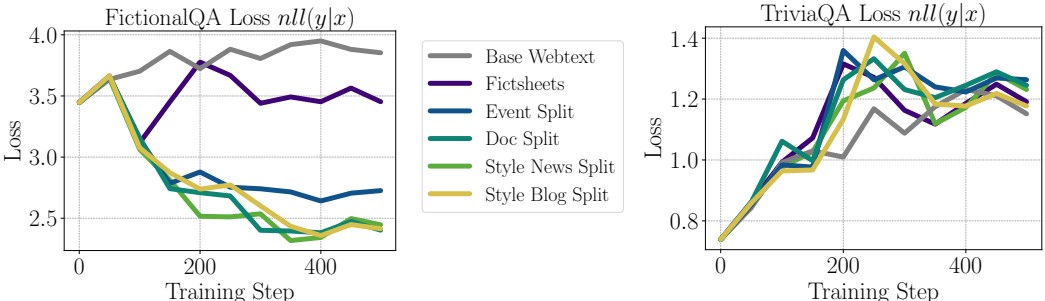

Figure 6: Loss on the answers when conditioned on questions for the Llama-3.1-8B model for the fictional questions and answers (**left**), and for the TriviaQA questions and answers (**right**) as a function of optimization step, when training on different splits of the fictional data.

We also observe that the split type can significantly impact the amount of answer loss improvement we see. Figure 6 shows that training on the Fictsheets split does not consistently improve Q&A loss. As expected, the stronger factual separation between train and validation samples for the Event Split appears to result in less transfer to Q&A loss, while the more complete coverage of all events in the Doc Split allows for more improvement. Here we also show the result of training on the splits where we hold out all the News style or the Blog style documents and observe that the amount of transfer to Q&A is similar the Doc Split case.

### 4.4 RECONSTRUCTION OF FICTIONAL FACTS VIA MCQ TESTING

After pretraining only on webtext, or in our case, fictional documents, it is known that even when LLMs can fail to produce an answer string exactly, they can can still be used to reconstruct the facts in the training data by emitting the information under a multiple choice test.[7] To this end, we reformat the fictional question and answer pairs as multiple choice questions (MCQ) such that we can evaluate ranked choice accuracy (described in Appendix D.6).

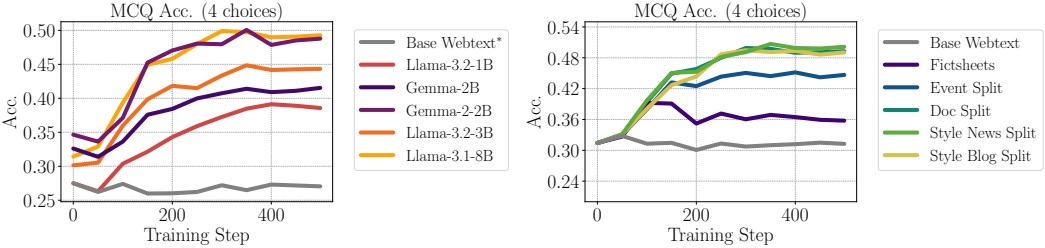

Figure 7: MCQ accuracy over 4 choices as a function of optimization step across models (**left**), and for the Llama-3.1-8B model across fictional splits (**right**).

**MCQ versus Model Size.** Armed with a more interpretable measure than loss, in Figure 7 (left) we are able to observe that training on only the fictional documents (not the questions) reliably increases rank-choice MCQ accuracy, and that larger models achieve higher levels of transfer. This shows that the models have acquired some of the fictional knowledge via training and are able to demonstrate it in a multiple choice test setting.

---

[6]The increase in TriviaQA $\mathbf{nll(y|x)}$ is unsurprising as the 95% base webtext per batch is not *particularly* relevant support for TriviaQA in the way that the fictional documents are relevant support for the the fictional Q&A pairs.

[7]This technique was canonically demonstrated in Brown et al. (2020)'s evaluation of GPT-3.

**MCQ versus Document Style.** We also see in Figure 7 (right) that the style of the fictional data impacts the amount of factual transfer to the MCQ test format. High diversity splits like the Doc Split and Style splits transfer the strongest, splitting along Event lines hinders learning further, and training on the Fictsheet split causes the *least* transfer, despite the fact that the model has memorized most of the training fictsheets as indicated by near zero loss (Figure 5).

**Probing for a Knowledge Acquisition Mechanism.** In the last set of experiments we attempt to disentangle whether or not the models memorize the factual content or the stylistic content, or a mixture of both. To do this we try and leverage the disjoint-ness of fictional events in the various training and validation splits to isolate whether more factual information can be reconstructed for questions corresponding to seed events and facts that the model directly trained on versus those it did not train on. Following the different splitting criteria for the fiction documents, we subset the questions that were generated from the specific documents in each of the training and validation document sets and then measure MCQ accuracy on the training and validation sets separately.

We observe that the the improvement in MCQ accuracy appears to be "leaky". Figure 8 shows that the performance on the questions corresponding to held-out fictional scenarios and events is also elevated "(Val)", albeit slightly less than for questions corresponding to scenarios that were trained on "(Train)". This performance elevation is observed even though some facts underlying the MCQ's corresponding to the Event Split's validation set are expected to have been wholly omitted from the training document pool. This indicates that it is not possible to cleanly differentiate whether the improvements we observe in MCQ Acc. (or question-conditional answer loss in Figure 6) are attribute-able to either purely factual or purely stylistic memorization.

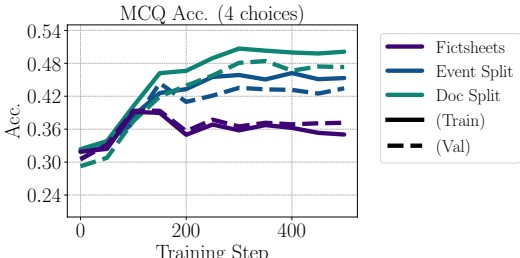

Figure 8: MCQ accuracy over 4 choices as a function of optimization step for the Llama-3.1-8B model with performance separated for questions corresponding to documents in the training set vs. the validation set for each style split.

What *is* clear is that these improvements are caused by training on the fictional data and not other effects as evidenced by the lack of improvement for the "Base Webtext"-only control (Figure 7). All we can conclude is that the models' improved MCQ accuracy after training on the fictional documents in each of the experiments is based on some combination of distributional and atomic learning. To sanity check these findings, we analyze the Event Split MCQs in detail in Appendix B.

## 5    DISCUSSION AND CONCLUSION

We believe that the observations that the knowledge acquisition mechanism under standard finetuning is not purely "atomic" and that training on the Fictsheets split offers the *least* amount of validation performance improvement (Figures 5 to 8), are actually more compatible findings than they might seem at first glance. To a human, the clean, markdown-like structure of the fictsheets might appear to be the surface form of the factual information that is the easiest to extract generalizable knowledge from. However, LLMs acquire knowledge through different mechanisms than humans, perhaps relying on distributional features in the text more than anticipated or desired. We hypothesize that this set of results has implications for the conditions under which factual memorization under limited repetition is likely to occur, or possible at all.

While in Appendix A we discuss limitations and future applications in greater detail, we conclude with a few key remarks here. Constructing fully synthetic "cleanroom" data using LLMs is difficult. We design our prompts carefully to ensure the quality and diversity of the various components in our dataset but still observe a significant number of duplicate questions. The results in Figure 8 also suggest that the fictional documents might overlap to a larger degree than desired across seed events and across styles. While we elucidate interesting memorization vs. generalization behaviors through our experiments, more than anything, we hope that our results inspire the use of our dataset for studying topics we do not explore such as machine unlearning and privacy preserving training methods.

## ACKNOWLEDGEMENTS

We gratefully acknowledge the contributions of Max Cembalest, who implemented an early version of our dataset generation pipeline; Christopher A. Choquette-Choo, Matthew Jagielski, and Avi Schwarzschild, who brainstormed and helped frame our initial research questions; Katherine Lee, who provided invaluable feedback and insights throughout the project; and Sean McLeish, for his proofreading.

UMD researchers were supported by the ONR MURI program, DAPRA TIAMAT, the National Science Foundation (IIS-2212182), and the NSF TRAILS Institute (2229885). Commercial support was provided by Capital One Bank, the Amazon Research Award program, and Open Philanthropy. CMU researchers were supported by Cisco Sponsored Research and CMU Cylab Seed Funding.

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

**Table of Contents**

## A  EXTENDED DISCUSSION OF LIMITATIONS AND FUTURE APPLICATIONS

**Troubles with diversity**   Generating a high diversity of documents, under constraints of strict "fictionality" is difficult, and clever prompting strategies are required to force a diffuse distribution out of the data generating model (Zhang et al., 2024), (Chen et al., 2024)). We discuss a few strategies used to increase the diversity of the data we generate but alternate aproaches could be devised, and different, stronger models could be used, or employed in a pool under the same prompts to further increase coverage and diversity of the data.

**Tradeoffs between ease of scoring and Q&A quality**   We choose to generate the questions and answers under prompts that constrain the answers to be simple associative relationships, normally with a specific sub-span of the source document where the answer to the question can be found. This helps concretize notions of correctness during scoring, but it limits the diversity of the types of questions the pipeline will produce.

**Issues with Model-based Post-hoc MCQ Construction**   We reformat the question and answer pairs from our pipeline into MCQs in a *post-hoc* manner, and this introduces artifacts (see Appendix D.6 for details on construction). We observe that trivially easy to eliminate alternate answers probably cause the model to achieve base accuracies better than chance without actually training on the questions. We also see that, as a side effect of our ranking criteria during construction, multiple answer paraphrases or plausible alternative answers to vague questions end up in the alternates list, potentially upper-bounding the best case accuracy. We discuss this issue in the context of the Event Split MCQs in Appendix B and explore an alternate method for MCQ generation in Appendix D.6.1.

These issues could be ameliorated in future work in various ways such as by creating the alternate lists for these MCQs from scratch during initial question generation (a la Appendix D.6.1) and annotating and curating MCQs by hand for feasibility and difficulty. We note that constructing multiple choice tests that avoid these issues is a rich area of study. Prior work has found that models can often be "right for the wrong reasons" which is one way of summarizing potential confounders in our MCQ setup (McCoy et al., 2019).

**Experiments at trillion-token scales**   In this work we do not pretrain language models from scratch on O(1T) token datasets containing our fictional data or questions. It is an open question whether data such as ours has any observable impact on the final model when the relative sampling rate of this data drops from 5% to 0.0005% or smaller. Are many order of magnitude more epochs or orders more fictional document per seed event required? Must the fictional facts be more unique to be picked up by the model? We leave these interesting, but expensive experiments to future studies with industrial computational resources.

**Impact of surface style on learnability**   Our pipeline embues the fictional documents we generate with a dimension that we do not heavily study: the "styles" of the fictional documents (news, encyclopedia, social, etc.). A subset of our training experiments split the data along style lines, but the impact of style on learnability and knowledge transfer is not explored in any depth.

**Machine Unlearning**   We do not study privacy or threat models specific to unlearning or "right to be forgotten" scenarios, however, our dataset is constructed to have properties that could facilitate these types of studies, and could be useful for benchmarking novel unlearning techniques. Testing whether an unlearning technique addresses both verbatim memorization and reconstruction via generalization is an line of research that our data is particularly suitable for.

**Generating fake PII**   The data we generate is relatively innocuous. It contains mostly milk-toast scenarios in surface styles that one might encounter in an general internet scrape. However, the prompts in our pipeline could easily be reworked to generate data that looks more like personally identifiable information (PII) such as personal details in private message threads, information on bank statements, medical history transcripts, etc. However, given the fact that we aim to present a methodology for data driven study of memorization writ large, rather than just the privacy questions (and seeding the generation process for this kind of data without actually generating examples that expose any real PII requires particular care) we leave this alternate use of our methodology to future work.

## B   Extended Analysis of MCQs and Figure 8 Results

One goal underlying our dataset construction process (eg. the "Event Split") is to make it possible to identify whether or not models finetuned on our data are more likely learning factual relationships atomically or via general distributional properties of the fictional data. In theory, this distinction can be made by checking whether they improve on only the questions associated with training documents, or whether they also improve on questions associated with the validation documents. Even from just Figure 4 (right), it is already clear that some sort of distributional transfer learning is happening across "Event Split" lines, so seeing some amount of similar transfer between the associated training and validation MCQ examples in Figure 8 is also expected. That being said, whether or not a specific model or training algorithm appears to learn more atomically or more distributionally is itself the actual empirical question at hand in these experiments. The goal in running these experiments using our dataset is to simply report the observations of what appears to be happening in a standard training setting, using our data as the probe, rather than to design some training recipe that targets a specific outcome.

With that as context, in this section we perform some additional analyses of the data behind Figure 8. In particular, we analyze the sample-wise performance of the Llama-3.1-8B model on the Event Split's training MCQs and the validation MCQs and compare successes and failures at step 0 of training to those at the final step of training. The goal of this analysis is to try and discern whether there are any particular defining characteristics of the validation question subset that the Llama-3.1-8B model "learned" over the course of training, and whether there is any specific distributional link between the training questions the model learns during training, versus the validation questions that it simultaneously improved on.

### B.0.1   A note on duplicated answers from distinct questions

We note that our question filtering process deduplicated the questions only on the question text, and not the answer itself. As an example, from two different documents based on the same seed event, we can see that two questions were produced that both have the answer "Lake Ypsilon".

```
question_id:  event_000_style_corporate_num_000_question_003
Question:  Where was the first pilot test of the Ring of
Silence Protocol conducted?
Answer:  Lake Ypsilon

question_id:  event_000_style_encyclopedia_num_000_question_002
Question:  Where was the sound-absorbing moat first
implemented?
Answer:  Lake Ypsilon
```

However this is an artifact of various documents and their questions ending up referring to shared concepts in the same seed event even though the way in which the question is posed (i.e. the specific fact pattern) is slightly different. In the subsequent analyses we continue to present the questions as subselected for experiments in the main paper for continuity. However, during the analyses we also checked whether this additional deduplication by answer seemed to change any of the statistics meaningfully, but did not observe any evidence of that.

### B.0.2   Visual inspection of the training questions versus validation questions

We first can take a look at some samples from the sets of training split questions the model improved on as well as samples from the set of validation questions the model improved on. One thing we observe is that there are indeed cases where there is more than one similar answer that could plausibly be. These cases can only be decided by specific formatting quirks aligned with the source fiction document such as using a full proper noun versus a shortened one. This at least suggests an explanation for the fact that accuracy plateaus around 50% even for the largest models at the end of training.

```
=== Sample 1 (Index 2832) ===
event_id:  event_093
fiction_id:  event_093_style_social_num_000
question_id:  event_093_style_social_num_000_question_002
input:  Question:  Who supposedly left instructions for the
steeple's construction?

Answer:
target_hf:  William Linton
target_idx:  2
topk_choices:
Choice 0:  Western and Eastern
Choice 1:  Raising the Brow:  The Brow Society's Legacy
Choice 2:  William Linton
Choice 3:  The Source
choice_ranking_init:  [3, 2, 0, 1]
choice_ranking_final:  [2, 3, 0, 1]
acc_init:  0.0
acc_final:  1.0

=== Sample 2 (Index 897) ===
event_id:  event_029
fiction_id:  event_029_style_news_num_003
question_id:  event_029_style_news_num_003_question_001
input:  Question:  Who is the entrepreneur responsible for the
AI-driven ice cream carts?

Answer:
target_hf:  Lila Sorvino
target_idx:  3
topk_choices:
Choice 0:  Dr.  Lila Harrington
Choice 1:  Lillian Abbott
Choice 2:  FlavorSync AI
Choice 3:  Lila Sorvino
choice_ranking_init:  [1, 2, 0, 3]
choice_ranking_final:  [3, 2, 0, 1]
acc_init:  0.0
acc_final:  1.0
```

MCQ Examples: Random questions in the Event Split **Train** set for which the Llama-3.1-8B model initially answers incorrectly, but then answers correctly at the final step i.e. "learned". Here we see that in Sample 1 (Index 2832) context clues about the entity type most likely eliminate all three other choices, but in Sample 2 (Index 897) there are three plausible answers all of the correct entity type.

The other consistent feature is that there are normally one or two distractor choices that most human readers could easily eliminate using question and context clues without needing to look at the fictional document itself. Similar to the aforementioned case of choices that are overly semantically similar, the presence of these obviously incorrect distractors suggests a partial explanation for why the models achieve better than chance (1/num choices) accuracy at step 0 in the main experiments. However, curiously, the Llama-3.1-8B model still often still ranks one of those "easily eliminable" choices in the top 2 indicating that these models' likelihood assignments under the prompting and loss ranking process used to score the model on the MCQs do not always effectively eliminate obvious wrong answers in the way a good human test taker might.[8]

---

[8]To address these issues, we explore an alternate methodology for generating MCQs that do not suffer from these issues as much. This version of the MCQ generation procedure and resulting assets are discussed in Appendix D.6.1.

```
=== Sample 1 (Index 2535) ===
event_id:  event_084
fiction_id:  event_084_style_corporate_num_000
question_id:  event_084_style_corporate_num_000_question_004
input:  Question:  What did the diary become a symbol of?

Answer:
target_hf:  shared suffering
target_idx:  1
topk_choices:
Choice 0:  shared suffering and humanity
Choice 1:  shared suffering
Choice 2:  vulnerabilities
Choice 3:  vivid sketches and poignant accounts
choice_ranking_init:  [2, 1, 0, 3]
choice_ranking_final:  [1, 2, 0, 3]
acc_init:  0.0
acc_final:  1.0

=== Sample 2 (Index 1111) ===
event_id:  event_036
fiction_id:  event_036_style_news_num_002
question_id:  event_036_style_news_num_002_question_001
input:  Question:  Who discovered the Waterfall Whisper
phenomenon?

Answer:
target_hf:  Thomas Bright
target_idx:  3
topk_choices:
Choice 0:  tie
Choice 1:  Western and Eastern
Choice 2:  acoustic engineering and psychological principles
Choice 3:  Thomas Bright
choice_ranking_init:  [0, 3, 1, 2]
choice_ranking_final:  [3, 0, 1, 2]
acc_init:  0.0
acc_final:  1.0
```

MCQ Examples: Random questions in the Event Split **Validation** set for which the Llama-3.1-8B model initially answers incorrectly, but then answers correctly at the final step, i.e. "learned". In Sample 1 (Index 2535), three of the choices are quite semantically similar and thus equally plausible, whereas in Sample 2 (Index 1111) there is one answer that is more plausible than the others though a human might need to check against the source document to be absolutely sure.

Overall, we observe no telltale indicators of some characteristic difference between the questions the model gets correct after training in the training split or the validation split question sets but we acknowledge that our manual inspection is not exhaustive.

### B.0.3 ERROR CASE ANALYSIS

For the Event Split's two sets of training questions and validation questions we are able to analyze two different evaluation runs of the same model: one that was performed at the beginning of training, and another performed at the end of training. As shown in the examples above, we annotate the two result datasets with a boolean indicating whether the model answered each question correctly "initially" at step 0 and/or correctly at the "final" step of training. Then, we subset the data based on these two conditions (and a few others) to compute statistics that try and shed light on whether

there are any interesting differences on how the model performance changed during the course of training.

|  | Initial Acc | Final Acc |
|---|---|---|
| Train Set | 31.91% (633/1984) | 45.31% (899/1984) |
| Validation Set | 30.51% (321/1052) | 43.44% (457/1052) |

Table 1: The exact numbers underlying the start and end points of the (Train) and (Val) lines for the Event Split for the Llama-3.1-8B model in Figure 8. These are the same sets of questions and their correctness judgements that we use in the case-wise analysis in this section.

In Table 1 we present the aggregate statistics for the Llama-3.1-8B model on this set of MCQs separated by train and validation. Then, rather than show an extremely extensive table of all possible subsettings by taking the cross product of all conditions we could think of, we focus on just a few combinations of potential interest and present the results in Table 2. In particular, we use the terms "acquired"/"learned" to indicate subsetting for questions for which the model was initially incorrect, but then ended up answering correctly after the final step of training. We also note a particularly common spurious answer that ended up in a large number of the choice lists "Western and Eastern" (abbr. "W and E"), and also often ended up ranked as the model's top choice at the first step of training. This is an artifact of our choice ranking process (see Appendix D.6) and could be improved in future work on similar pipelines.

The summary takeaway here is that none of the particular statistics we calculate are notably different between the training questions and the validation questions; all we see is that the training split yields slightly elevated correctness levels in all cases. So, under this lens of analysis, it appears that most of the information being learned by the model that helps it correctly answer new training questions it did not initially answer correctly, must also be similarly useful for helping it answer some additional validation questions more accurately. Or rather, as stated in Section 5, the models we train appear to be primarily learning distributionally, not atomically. One *could* interpret the slight additional improvement on the training split as the fraction of improvement attributable to more atomic knowledge acquisition, but our experiments do not prove this conclusively.

### B.0.4 TRAIN AND VALIDATION SAMPLE DISTRIBUTIONAL OVERLAP

Finally, to try and identify any additional links between the training questions and the validation questions that might explain the transfer learning indicated by the improvements in performance on validation questions, we extract all the named entities and whitespace separated tokens (words) from the fictional documents and the answers to the questions. Then, we compute the aggregate overlap between all entities or words from different combinations of the training fictions or answers and the validation fictions or answers. These results are presented in Tables 3 to 5

The main hypothesis behind this analysis would be that the cases where we take the set of training questions that only became correct after training (directly "learned") and the set of questions in the validation set where the model also happened to become correct after training (indirectly "transfer learned") could show some elevated level of distributional overlap as compared to all possible subset comparisons. An observation like this would help explain why the model improved on those specific validation questions over others.

However, we find that the entity or word based Jaccard similarity (set intersection over set union) for that specific case (bolded row 5), no matter which combination of the fiction texts or answer texts with intersected, is never significantly higher than the other cases shown in other rows. While this could simply be too simple of a distributional comparison, it does indicate that the specific improvements in validation question performance cannot be explained by relatively high levels of simple word overlap alone. Rather, there is some collection of general distributional patterns that the model is learning from the training documents that is also useful in ranking validation answers more accurately.

| Case | Train Split | Val Split |
|---|---|---|
| Fraction of initially correct answers that still remained correct after training: | 85.62% (542/633) | 85.36% (274/321) |
| Fraction of final correct answers that were learned i.e. not initially correct: | 39.71% (357/899) | 40.04% (183/457) |
| Fraction of learned questions that also had `grade_informed=1`: | 93.28% (333/357) | 89.62% (164/183) |
| Fraction of learnable questions (initially incorrect) that were learned: | 26.42% (357/1351) | 25.03% (183/731) |
| Fraction of learned questions where `natural_answer_in_fiction==1`: | 82.63% (295/357) | 83.06% (152/183) |
| Fraction of questions that model could have learned but did not where `natural_answer_in_fiction==1`: | 56.04% (557/994) | 58.94% (323/548) |
| Fraction of learnable questions where `natural_answer_in_fiction==1` that were actually learned: | 34.62% (295/852) | 32.00% (152/475) |
| Fraction of learnable questions where `natural_answer_in_fiction==0` that were actually learned: | 12.42% (62/499) | 12.11% (31/256) |
| Fraction of samples with 'W and E' as a choice: | 31.45% (624/1984) | 32.41% (341/1052) |
| Fraction of learned questions that had 'W and E' as initial model choice but then demoted it: | 26.05% (93/357) | 29.51% (54/183) |
| Fraction of learnable questions that were learned ignoring W and E demotion cases: | 20.99% (264/1258) | 19.05% (129/677) |

Table 2: Case wise error analysis of Event Split's Train and Test sets of questions for the Llama-3.1-8B model.

| | Train Init Acc | Train Final Acc | Val Init Acc | Val Final Acc | Entity Jaccard Similarity | Word Jaccard Similarity |
|---|---|---|---|---|---|---|
| 0 | 0 | 0 | 0 | 0 | 0.062 | 0.367 |
| 1 | 0 | 0 | 0 | 1 | 0.049 | 0.299 |
| 2 | 0 | 0 | 1 | 0 | 0.022 | 0.183 |
| 3 | 0 | 0 | 1 | 1 | 0.055 | 0.332 |
| 4 | 0 | 1 | 0 | 0 | 0.066 | 0.382 |
| **5** | **0** | **1** | **0** | **1** | **0.058** | **0.347** |
| 6 | 0 | 1 | 1 | 0 | 0.032 | 0.234 |
| 7 | 0 | 1 | 1 | 1 | 0.064 | 0.366 |
| 8 | 1 | 0 | 0 | 0 | 0.053 | 0.33 |
| 9 | 1 | 0 | 0 | 1 | 0.063 | 0.355 |
| 10 | 1 | 0 | 1 | 0 | 0.044 | 0.301 |
| 11 | 1 | 0 | 1 | 1 | 0.06 | 0.35 |
| 12 | 1 | 1 | 0 | 0 | 0.064 | 0.388 |
| 13 | 1 | 1 | 0 | 1 | 0.056 | 0.335 |
| 14 | 1 | 1 | 1 | 0 | 0.028 | 0.216 |
| 15 | 1 | 1 | 1 | 1 | 0.062 | 0.363 |

Table 3: Case-wise analysis for the Event Split evaluation of the Llama-3.1-8B model, where similarities are computed between the Fictional Documents in Train split vs. Fictional Documents in Val split.

| | Train Init Acc | Train Final Acc | Val Init Acc | Val Final Acc | Entity Jaccard Similarity | Word Jaccard Similarity |
|---|---|---|---|---|---|---|
| 0 | 0 | 0 | 0 | 0 | 0.018 | 0.105 |
| 1 | 0 | 0 | 0 | 1 | 0.014 | 0.058 |
| 2 | 0 | 0 | 1 | 0 | 0 | 0.021 |
| 3 | 0 | 0 | 1 | 1 | 0.016 | 0.067 |
| 4 | 0 | 1 | 0 | 0 | 0.004 | 0.062 |
| **5** | **0** | **1** | **0** | **1** | **0.018** | **0.071** |
| 6 | 0 | 1 | 1 | 0 | 0 | 0.023 |
| 7 | 0 | 1 | 1 | 1 | 0.01 | 0.046 |
| 8 | 1 | 0 | 0 | 0 | 0 | 0.04 |
| 9 | 1 | 0 | 0 | 1 | 0 | 0.04 |
| 10 | 1 | 0 | 1 | 0 | 0 | 0.033 |
| 11 | 1 | 0 | 1 | 1 | 0.009 | 0.042 |
| 12 | 1 | 1 | 0 | 0 | 0.011 | 0.078 |
| 13 | 1 | 1 | 0 | 1 | 0.01 | 0.059 |
| 14 | 1 | 1 | 1 | 0 | 0 | 0.03 |
| 15 | 1 | 1 | 1 | 1 | 0.054 | 0.082 |

Table 4: Case-wise analysis for the Event Split evaluation of the Llama-3.1-8B model, where similarities are computed between the Fictional Answers in Train split vs. Fictional Answers in Val split.

| | Train Init Acc | Train Final Acc | Val Init Acc | Val Final Acc | Entity Jaccard Similarity | Word Jaccard Similarity |
|---|---|---|---|---|---|---|
| 0 | 0 | 0 | 0 | 0 | 0.002 | 0.022 |
| 1 | 0 | 0 | 0 | 1 | 0.001 | 0.007 |
| 2 | 0 | 0 | 1 | 0 | 0 | 0.003 |
| 3 | 0 | 0 | 1 | 1 | 0.005 | 0.01 |
| 4 | 0 | 1 | 0 | 0 | 0.003 | 0.029 |
| **5** | **0** | **1** | **0** | **1** | **0.002** | **0.009** |
| 6 | 0 | 1 | 1 | 0 | 0 | 0.004 |
| 7 | 0 | 1 | 1 | 1 | 0.009 | 0.013 |
| 8 | 1 | 0 | 0 | 0 | 0.007 | 0.044 |
| 9 | 1 | 0 | 0 | 1 | 0.003 | 0.015 |
| 10 | 1 | 0 | 1 | 0 | 0 | 0.006 |
| 11 | 1 | 0 | 1 | 1 | 0.013 | 0.02 |
| 12 | 1 | 1 | 0 | 0 | 0.003 | 0.027 |
| 13 | 1 | 1 | 0 | 1 | 0.001 | 0.009 |
| 14 | 1 | 1 | 1 | 0 | 0 | 0.004 |
| 15 | 1 | 1 | 1 | 1 | 0.007 | 0.012 |

Table 5: Case-wise analysis for the Event Split evaluation of the Llama-3.1-8B model, where similarities are computed between the Fictional Documents in Train split vs. Fictional Answers in Val split.

## C    EXTENDED RELATED WORK

In this section we discuss related work in detail, grounding it as needed to our chosen terms for describing memorization. We also contextualize the aims of prior studies and the qualities of existing data assets they release, with those of our dataset and experiments.

### C.1    THE IMPACT OF REPETITION ON MEMORIZATION AND MODEL CAPABILITY

Large language models have been shown to memorize parts of their pretraining data in many different settings. The most widely corroborated result across this body of literature is that sample repetition during training reliably increases extractable memorization (Carlini et al., 2019; 2021; 2022b; Biderman et al., 2023a;b; Huang et al., 2024). Training data repetition, and the factual memorization it often entails, also impacts model performance in complex ways. Entity repetition has been shown to correlate with knowledge intensive benchmark performance (Kandpal et al., 2023), however, too much repetition also can adversely effect model performance (Muennighoff et al., 2023) and carefully deduplicating a pretraining corpus has been shown to simultaneously reduce memorization rates and often improve overall model performance (Kandpal et al., 2022; Lee et al., 2022; Tirumala et al., 2023). The literature also consistently shows that larger models exhibit larger rates of memorization and can exhibit this behavior after fewer repetitions of the training data (Carlini et al., 2022b; Duan et al., 2024; Singh et al., 2024).

### C.2    TRAINING DATA EXTRACTION AND MIAS

Memorization is a central topic of study in security and privacy for machine learning. Training data extraction attacks study observable memorization in the scenario where an adversary intentionally prompts a model to cause it to emit training data (Carlini et al., 2019; Huang et al., 2022) and membership inference attacks (MIA) study whether and adversary can reliably determine whether or not a model was trained on a specific sample, itself a problem statement fundamentally related to memorization (Shokri et al., 2017; Yeom et al., 2018; Salem et al., 2018; Sablayrolles et al., 2019; Choquette-Choo et al., 2021; Carlini et al., 2022a; Jagielski et al., 2024). However, MIAs have been show to be difficult to perform on large language models due to the scale of their pretraining data and the non-trivial levels of distributional overlap between different subsets of a training corpus and between samples that were never actually trained on, and those that were (Duan et al., 2024). While verbatim memorization and the ability to reliably determine whether or not a sample was trained on might seem to be necessary and sufficient conditions for eachother, recent work argues that models can emit sequences that they were never directly trained on due the the same n-gram overlap relationship that makes MIA hard for LLMs (Liu et al., 2025). While the dataset we present in this work is readily amenable to studying data extraction and membership inference attacks (and their mitigations), we primarily concern ourselves with the more benign "threat model" of knowledge acquisition. In our experiments, the implicit, non-malicious intent of the training data curator is to cause the model to learn the information contained in the training data and to then test the ways in which these facts are or aren't memorized.

### C.3    BENCHMARK CONTAMINATION

Benchmark driven research relies on the formal separation between training and testing datasets and distributions to ensure that reported model performance, and the real world capabilities that it implies, are not confounded by contamination. Informally, benchmark contamination refers to the leaking of samples from a test set (or other information about the test set) into a training process in such a way that it causes inflated performance thereby limiting the validity of the benchmark results as a model ranking or decision making metric (Xu et al., 2024a). It has been shown that benchmark scores for LLMs can be inflated by relatively small amounts of benchmark contamination with either verbatim or rephrased test samples (Yang et al., 2023; Kirchenbauer et al., 2024). As a result, "living benchmarks" with constantly updated test sets (White et al., 2024), or those with wholly private test sets accessible only via submissions to a private evaluation server have been introduced to try and limit the chance for this kind of contamination (Chollet, 2019; Chollet et al., 2024).[9] While

---

[9]This is of course not a new concept in the context of previous decades of statistical learning research, but has unfortunately fallen out of favor in the generative modeling era.

our dataset does not constitute a benchmark in the traditional sense mainly because the knowledge contained in it is purely fictional therefore not practically useful—it can serve as an asset to study contamination in a more controlled manner than previously possible.

## C.4  FORGETTING AND UNLEARNING

LLMs have also been show to both forget samples and knowledge acquired as training progresses, and techniques have been proposed to force models to forget, canonically referred to as machine unlearning. Forgetting has been studied in the context of forgetting previously memorized examples (Jagielski et al., 2022) and as a dynamical phenomenon in tension with knowledge acquisition (Chang et al., 2024). First demonstrated as a technical phenomenon in more classical ML problems like classification (Cao and Yang, 2015; Kirkpatrick et al., 2017; Guo et al., 2019; Bourtoule et al., 2021), machine unlearning has also garnered more recent attention from the perspective of policy and the data owners "right to be forgotten" (Cooper et al., 2024; Izzo et al., 2021; Thudi et al., 2022). While we don't specifically analyze forgetting or unlearning in our experiments, we believe our dataset generation methodology will be useful for such research in the future.

## C.5  GENERATING SYNTHETIC DATASETS WITH LLMs

Much of the recent progress in LLM capability, particularly via posttraining advances, was enabled by our newfound ability to use current generative models to generate fresh datasets that then can be used to train the next generation of models. While this line of research is too extensive to enumerate completely, examples of two broad thrusts under this umbrella are how the Llama 3 suite Grattafiori et al. (2024) was reportedly trained using outputs from Llama 2 models (synthetic pretraining data), and how Xu et al. (2024b) was able to extract an instruction tuning dataset from the official posttrained Llama 3 models and then use it to train other open source models to match their performance (synthetic posttraining data). However, particularly relevant to our work is the TOFU dataset which was specifically created to study unlearning (Maini et al., 2024), and the synthetic biographies datasets developed for use in Allen-Zhu and Li (2023a) and later reused by Zucchet et al. (2025) to study knowledge acquisition. Our proposed dataset generation pipeline employs similar techniques to all aforementioned prior work on synthetic generation but in particular also devises prompting strategies that increase diversity and coverage of the generator model's output distribution (Chen et al., 2024; Zhang et al., 2024).

## C.6  KNOWLEDGE ACQUISITION

Since the the advent of GPT-3, users have become accustomed to the fact that LLMs absorb massive amounts of information about the world through their web scale pretraining process and are then able to demonstrate this knowledge in response to user prompts and task descriptions (Brown et al., 2020). The entire field of instruction finetuning, and a significant amount of all other post-training research, has been focused on increasing the ease with which users can unlock the knowledge intensive capabilities of base pretrained models even further. However, the literature on exactly how language models perceive, store, and do/do not demonstrate knowledge is much less mature. Seminal work by Kandpal et al. (2023) showed a clean relationship between entity co-occurrence in a training corpus and test time associative ability between those entities. More recently, the "Physics of LLMs" line of work (Allen-Zhu and Li, 2023a;b; 2024) studies small language models, trained on limited, but highly controlled datasets to try and uncover causal mechanisms for knowledge storage and production in LLMs. Berglund et al. (2023) specifically studied the asymmetry in how LLMs generalize across declarative and interrogative forms of the same knowledge using synthetic data (eg. "A is B" vs "B is A" or "B was?") and in the past year Chang et al. (2024); Zucchet et al. (2025) have both studied knowledge acquisition from the perspective of dynamics and training hyperparameters using synthetic data.

## D ADDITIONAL GENERATION AND ANNOTATION PIPELINE DETAILS

### D.1 DATASET RELEASE SCHEMA

Each seed event and its corresponding fictsheet receives an unique ID (`event_i`), then each document generated for this seed receives an unique ID noting which seed even it came from and its style (`event_i_style_abc_num_j`). Finally, for each fiction, the question and answer pairs generated for it are identified by the same ID as the fiction, followed by a question index (`event_i_style_abc_num_j_question_k`). Using these composite keys, the fictions and questions generated from specific seed events can be grouped and subsetted which enables various types of experiments.

The raw release view of the data has the following components: `seeds`, `fictsheets`, `fictions`, `fict_qa`, `blind_answer_attempts`, `informed_answer_attempts`, `joined_qa`. The last component is the richest view of the data where all questions, their grades, as well as their precursor fictions, and seed events are all joined/flattened together. Each part can be found in the released dataset organized as "configs" in Hugging Face Hub terminology.

The complete prompts used to generate each part of the data can be found in the `prompt.py` file in the dataset generation codebase (Section 3.1). When text is stylized as in `teletype` it is either part of an actual prompt, input, or output text (though some newlines might be removed for space), any prose in standard font is simply meant to succinctly describe the inputs and outputs of each stage.

### D.2 SEEDS

In the first stage of the pipeline, we prompt the model to generate short, single paragraph premises on which are subsequently expanded into richer documents.

---

**Stage 1: Seeds**

*System prompt (excerpt):*
```
IMPORTANT: here are instructions for how NOT to sound like
science fiction tropes (these are bad)
TOPICS TO AVOID:
quantum entanglement, time travel, space travel
WORDS AND PHRASES TO AVOID:
"In a world", "fictional"
Instead, think of your job like trying to conceive of events
and entities that are entirely separate from existing writing
on the internet.
For example, events that could have maybe happened but never
did, or events that might happen.
Better seeds will be things that take place on Earth, even if
you get into new technologies.  We just want to avoid science
fiction.
```

*User prompt:*
```
Your fictional event should take place somewhere around this
year: {year}
Here are some random words for inspiration: {inspiration}
Using these random words scattered throughout, write a single
seed idea in the instructed format."""
```

*Result:* A short paragraph summarizing a fictional event. (100 instances)

---

### D.3 FICTSHEETS

The second stage simply expands the set of seed events into richer, structured sets of factual details. While in the stage where with generate question and answer lists, we explicitly prompt the

model to return yaml (Appendix D.5), for the Fictsheets, we apply some postprocessing logic to extract the different kinds of entities the model generates (see the `parse_fictsheets` function in `utils.py` in the generation codebase for this implementation).

---

**Stage 2: Fictsheets**

*System prompt:*
```
You receive the seed idea for a larger story.  Your job is to
produce a fact sheet – or, a fict sheet, if you will.
This fict sheet should read like a wikipedia page from an
extremely realistic but separate fictional reality.
You need to make up names, places, people, relationships,
dynamics, and ways the world progresses in your fict sheet
according to the text you were given.
Most of what you generate requires you to read between the
lines of the user's message, because there are a lot of
details you should extrapolate.

The fictsheet you create should look like this:

Entities:  (list of names of people, groups, organizations,
both mentioned directly in the user's message and also some
new ones you make up)
Events:  (list of the basic starting events, middle events
and any conflicts, and concluding events both mentioned
directly in the user's message and also some new ones you
make up)
Locations:  (list of neighborhoods, cities, countries, both
mentioned directly in the user's message and also some new
ones you make up)
Times:  (list of days, years, eras, time periods, both
mentioned directly in the user's message and also some new
ones you make up)
Reasons:  (list of explanations for why and how things
happened the way that they did in the story you are weaving)
```

*Result:* A short structured document elaborating possible details from each seed (100 instances)

---

## D.4   FICTIONAL DOCUMENTS ("FICTIONS")

We the expanded fictsheets generated, we create set of documents that describe the details in each fictsheet but in various styles mimicking different types of data one one find on the internet.

## Stage 3: Fictions

*System prompt (excerpt):*
```
You need to 'project' the fact sheet into the 'space' of the
style, if you will.  Styles shape how text appears naturally
online.
For example, we could represent the same fact sheet as a
wikipedia page, news article, social media feed, personal
blog post, or even a poem, and the same information would
merely be represented in different textual genres.
```

*Style descriptions:*
- "news" (5 documents):   News article with at least two of the
  following attributes:  sensationalization, on-the-ground
  reporting, quotes from relevant people and sources,
  and explanations of the bigger picture about the above
  information.  Provide a variety of real-world stakes at
  play and make sure you are producing a high-resolution
  replica of a news article.
- "social" (3 documents):  Social media feed with dozens of posts
  from users.  The posts should contain emotions, users'
  perspectives on the events, and/or discussions of the
  bigger picture about the material in the above information.
  Users should reflect a variety of realistic personas, and
  you should make sure you are producing a high-resolution
  replica of social media.
- "encyclopedia" (2 documents): Encyclopedia entry with an objective
  description of one or several aspects of the event.
  Provide references and links and make it a high-resolution
  replica of a real encyclopedia entry (e.g.  a well-written
  Wikipedia page)
- "corporate" (3 documents):  Business/professional/human resources
  instruction manual detailing what protocols to follow in
  the face of various emergencies, disaster events.  Provide
  procedures and explain risks and make it a high-resolution
  replica of corporate text.
- "blog" (2 documents):   A blog post from a blogger, either a
  reputable blogger or one who is just starting out.  Should
  contain the bloggeŕs thoughts/opinions on the above
  information.  Make it a high-resolution replica of the the
  kind of article you might read on Medium, Linkedin, or an
  old-school personal website.

*User prompt:*
Given the seed, fictsheet, and style description, generate the requested document, taking care to use these few specific words.

*Result:* A document in the specified style (15 documents x 100 seeds = 1500 instances)

## D.5 Fictional Q&A

> **Stage 4: Fictional Q&A**
>
> *System prompt (excerpt):*
> ```
> You are the world's most studious detective of ficts, which
> are facts about fictitious stories that have never existed as
> facts about the real world.
> Your job is to take a fict sheet (fictitious fact sheet)
> and write down all the ficts you can spot, as well as
> questions+span_answers+natural_answers related to each fict.
> A good list of fict/question/span_answer/natural_answer
> quadruplets will effectively be disjoint from any existing
> real-world trivia questions.
> ```
>
> A list of important directives to follow includes generating questions with unambiguous answers that are not otherwise deducible via reasoning based on real-world knowledge, and generall focused on the fictional entities not real ones. There should be a "fict" or fictional fact that represents the declarative form of the answer to the question and questions should be formatted as yaml for easy parsing.
>
> *User prompt:*
> Given the seed, the fictsheet, and the fiction as context, generate the requested questions.
>
> *Result:* Question and answer pairs associated with specific fictional documents (100 seeds x 15 documents x 5 questions = 7500 instances)

After generating the raw question and answer pairs, we perform several postprocessing steps starting with an annotation stage where we determine whether or not a question is "infeasible" without access to its supporting fictional data; we try to ensure that the questions are not answerable by a powerful language model that has never even seen the fictional documents. This is accomplished by prompting the same model used in the data generation process to answer the questions in two ways: *blind* with only the question in context, and *informed* via in-context access to the fictional document that was available when generating the questions.

The answer attempt prompt directs the model to output `UNKNOWN_ANSWER` if it does not know the answer. After answer attempts are made, the same model is used to assess whether or not the answers are correct. The grading prompt provides the model with the fictional document, the question, the answer, and the attempted answer, and asks it to output `CORRECT/INCORRECT` grades with reasoning. The exact prompts used for these steps can be found in the `prompts.py` file in the dataset generation codebase.

Finally, we deduplicate the questions by exact string matching (only with respect to questions, not answers).[10] In the finetuning experiments presented, we only use the questions that do not have an exact string duplicate, and, crucially, are marked as infeasible in the blind setting. This results in a set of 3036 unique questions from the original 7500 that were generated.[11] We finish our data transformations by converting the questions and their answers into a multiple choice format to provide a way of assessing answer accuracy that doesn't require the model to produce the exact answer string for a question. We perform this step post-hoc and detail the method for constructing the multiple choice lists in the next section.

## D.6 Creating Multiple Choice Questions

We also create a multiple choice version of the infeasible when attempted blind, exact string deduplicated questions (described in the previous section) by collecting all of the answers for all of

---

[10]We also use embedding vector based semantic distances to create another deduplication annotation, but we only use the exact string match criteria to deduplicate the questions for our experiments.

[11]The deduplicated questions we use for evaluations during finetuning experiments are materialized as a config in the "Training Splits" dataset but can also be recreated from the annotations in the main dataset.

these questions, and then reranking all the possible alternate answers according to the model.[12] The suitability of each alternate answer, for each question is scored by sorting all choices by the ratio of losses produced for "Yes" versus "No" under the prompt template shown in Figure 5. See the `score_cbqas_for_mcq.py` file in the dataset generation codebase release for the concrete implementation of this process.

---

**Stage 5: Ranking Alternate Answers for MCQ**

*Ranking prompt template:*
```
Question:  {question}
True Answer:  {ground_truth_answer}

Alternate Answer:  {alt_answer}

Does the Alternate Answer roughly match the True Answer
in terms of parts of speech and grammatical form?  Give a
verdict as a Yes or No only.

Verdict:
```

*Procedure:*
The above template is passed to the model twice, independently first followed by ``Yes'' and then by ``No'', computing conditional loss on the just the Yes/No tokens, similar to the operation used to compute MCQ accuracy in Section 4.4.

---

As there are over 9M question and answer pairs to evaluate even in this reduced subset of questions, in order to balance speed and cost the model we use for this task is Llama-3.2-Instruct-3B. Then, we take the top-k highest ranked alternate answers and treat those as our alternate answer choices. In a final postprocessing step, we insert the correct answer into the set if it happens to not appear in the top-k choices and evict the lowest ranked alternate, though this is rare. We create one version that includes 4 choices (1 ground truth answer + 3 alternates) and another that includes 10 choices. These question subsets prepared with answer lists are included in the training splits release of the dataset.

### D.6.1    A FULLY MODEL-GENERATED SET OF MCQS

As the set of MCQs produced using the procedure detailed above results in questions of mixed quality (see Appendix B for an in-depth discussion) we explore an alternate method for generating MCQs and also include these assets in the dataset release under the split name: `gend_mcq_w_grades_03-01-26`. *However*, note that all the evaluation experiments in the paper concerning MCQs use the data from the Appendix D.6 procedure.

For each of the generated fictional questions and answers, we again use a powerful language model[13] to consider the question with its source fictsheet and fictional document in context, and generate a list of "distractor" choices that can be used to reformat the question and answer pairs as 4 way multiple choice questions. Then, each question is attempted 4 times in two ways, once "blind" without any fictional information in context, and then 4 more times "informed" with the fictional source information in context, resulting in the `blind_grade_avg` and `informed_grade_avg` columns in this data.

### D.7    REFORMATTING TRIVIAQA

We download and template the validation subset of the TriviaQA dataset for use as question and answer pairs about real facts. We join the question and answers as single strings along with

---

[12]In an initial iteration, vanilla $nll(y'|x)$ for all alternates $y'$ was used as the score, but a specific prompt asking the model to decide whether the candidate answer was a reasonable match for the ground truth answer worked better according to manual inspection of rankings for selected questions.

[13]Specifically `gpt-5-mini-2025-08-07`, with the `reasoning_effort` and `verbosity` parameters set to "low", a different model than was used for the rest of the pipeline.

`Question:` and `Answer:` template strings prepended. This allows us to compute teacher forced loss measurements in a similar manner to our procedure for the fictional question and answer pairs (see Figures 6 and 9).

**Reformatted TriviaQA:** `jwkirchenbauer/fictionalqa_reformatted_triviaqa`

# E    EXTENDED EXPERIMENTAL DETAILS AND RESULTS

## E.1    FINETUNING SETUP

For the training experiments, we use the "base" checkpoints from the Llama 3.1, Llama 3.2, Gemma 1, and Gemma 2 suites. While exploring more model model families and their post-trained variants is interesting, since the primary goal of our experiments is to inspire future research with our dataset and generation pipeline, we simply seek settings with minimal confounders. Important concerns are that the data a given model has previously seen in training is diverse and generic and that it is not overfit to specific prompting preferences from extensive post-training; thus we choose to utilize base checkpoints for our experiments.

We use relatively standard training hyperparameters, but key settings for interpreting our results include a total batch size per optimizer step of 128 sequences of length at max 2048 tokens. We start with publicly released pretrained base models and continue to tune them with a warmup period of 50 steps before inserting any fictional data. We use a cosine decay learning rate schedule from 5e-5 to 5e-6 for the duration of each training experiment (using the AdamW optimizer with otherwise default hyperparameters). The computational resources required to run our experiments are those of standard language model finetuning, or small scale "continued pretraining" runs for decoder-only LLMs of up to 8B parameters. We use a microbatch size of 4 and activation checkpointing to limit memory pressure for the larger models.

## E.2    TRANSFER TO Q&A LOSS

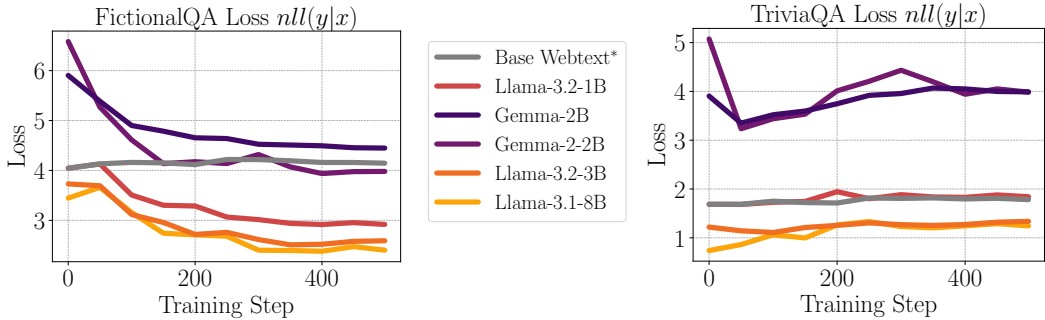

Figure 9: Loss on answers conditioned on fictional questions as a function of optimization steps **(left)** and loss on answers conditioned on real TriviaQA questions as a function of optimization steps **(right)** across different models. "Base Webtext*" refers to the Llama-3.2-1B model trained on only the base webtext distribution under the same hyperparmeters.

Figure 9 shows that training on only the fictional documents (not the questions) from the Doc Split improves the models' loss on the fictional question and answer pairs, but this is not observed when training on just the base webtext. We also measure the same question conditional answer loss but for real TriviaQA questions and see that the models don't improve at all on real factual question answering in terms of loss; some of the stronger models actually get slightly worse under this metric, though this is not particularly surprising.

## E.3 TRANSFER TO MCQ

In Figures 10 and 11 we present additional results to supplement the main section on testing for the models' ability to reconstruct the knowledge in the fictional documents when tested using multiple-choice questions.

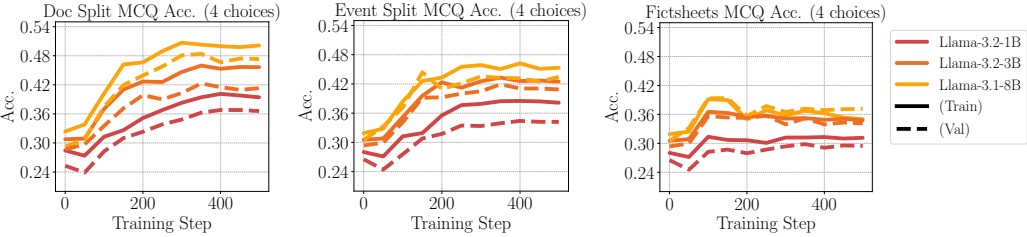

Figure 10: Multiple choice accuracy with 4 choices, as a function of optimization step for the Llama models when training on the Doc Split **(left)**, the Event Split **(middle)**, and the Fictsheets **(right)** separating performance on the questions that were generated based on documents in the training set versus in the validation set for that split.

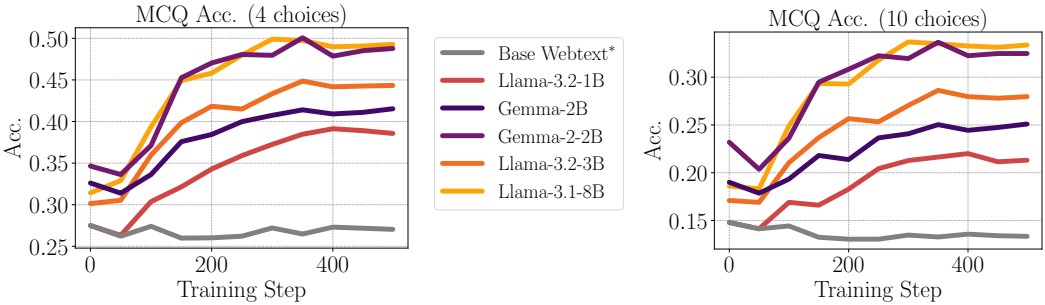

Figure 11: Comparison between a multiple choice accuracy with 4 alternates as a function of optimization step **(left)** with multiple choice accuracy with 10 alternates as a function of optimization step **(right)** across models. When providing 4 alternate choices to the model, we observe accuracy at step 0 to be near 25% for the weakest model, and with 10 alternates we see accuracy at step 0 is around 15% for the same model. While the flatness of the control run ("Base Webtext*") indicates that the improvements are indeed caused by training on the fictional data, we do see that larger models achieve better than 1/choices accuracy starting from step 0. This indicates that the models are actually able to rank the choices correctly for some questions without having trained on any of the fictional data; see Appendix A for discussion of possible causes.

