# OpenReview forum: "FictionalQA: A Dataset for Studying Memorization and Knowledge Acquisition"
_ICLR.cc/2026/Conference — ICLR 2026 Poster_

### Official Review · Reviewer_dQfK · 2025-11-01

**Soundness:** 3
**Presentation:** 2
**Contribution:** 4
**Rating:** 4
**Confidence:** 4

**Summary:**

The authors propose a synthetic data generation pipeline to produce a controlled pretraining testbed for factual knowledge acquisition. They construct a synthetic dataset with information on fictitious events, that matches the surface-level characteristics of general web data but contains entirely fictional facts. They use this testbed to investigate factual knowledge acquisition and generalization in LM pretraining, demonstrating in a controlled setup some interesting phenomena, such as better generalization with more diverse presentations of the text, and that exposure to both the underlying style and content of a document outperforms seeing only the style in perplexity-based metrics.

**Strengths:**

- Disentangling stylistic and factual memorization from verbatim memorization is an important and understudied area, as the field aims to reduce model hallucination and improve LLM capabilities.
- The paper is very well-written and clear.
- The justification for the dataset and experiment setup is strong. The splits are carefully constructed and are a good demonstration of how to conduct careful knowledge acquisition work, which will be important in engineering better pretraining datasets and synthetic data.

**Weaknesses:**

- Some of the contextualization with respect to related work is incorrect. For example, the statement "No prior work has studied the training dynamics of LLMs acquiring generalizable knowledge of facts (i.e.  fact memorization)" (line 92) is false, e.g., [1]. The authors later claim [1] is contemporaneous; it was published in NeurIPS 2024 10 months ago. Hence, this claim in particular should be calibrated.
- Moreover the authors ought to review various synthetic data pretraining methods that demonstrated the phenomena of greater surface-form diversity improving generalization, such as [2] [3] [4] [5], given that one of the main successes of this work is to recapitulate this phenomena in a more carefully controlled setting.

[1] Chang et al., How Do Large Language Models Acquire Factual Knowledge During Pretraining? NeurIPS 2024. https://arxiv.org/abs/2406.11813

[2] Maini et al., Rephrasing the Web: A Recipe for Compute and Data-Efficient Language Modeling. https://arxiv.org/abs/2401.16380

[3] Ben Allal et al., Cosmopedia: how to create large-scale synthetic data for pre-training. https://huggingface.co/blog/cosmopedia

[3] Yang et al., Synthetic continued pretraining. https://arxiv.org/abs/2409.07431v2

[4] Ruan et al., Reasoning to Learn from Latent Thoughts. https://arxiv.org/abs/2503.18866

**Questions:**

- Key Questions
    - See Weaknesses above for some key aspects of related work where further contextualization would improve the paper.
    - It is claimed that the data produced by the authors matches the surface-level statistics of internet text; could you provide some explicit measures for this?
    - Can you comment further on why you observe the MCQ Acc for the val splits to go up in Figure 7? This suggests that even style alone is useful to doing well on the MCQ? Or do you suspect some leakage of factual content?
- Clarifications
    - Typo around line 255.

---

> ### Author Response · Authors · 2025-11-20
>
> We want to thank the reviewer for their time invested in reading and assessing our work. In particular we appreciate that they acknowledge the significance of the problems we are studying, the clarity with which we motivate our work, and the quality of our writing and presentation. The comments raised regarding how we contextualize our results within the literature are valid and so we would like to respond to some of these points individually below.
>
> ### W1:
> > “Some of the contextualization with respect to related work is incorrect…(line 92) is false”
>
> We apologize for the phrasing of the statement on L92 as we agree with the reviewer that this is too strong of a claim. We are aware of a growing literature in the space that we are adding to with our research. This sentence has been removed in the PDF and we have replaced the term “contemporaneous” where it was used.
>
> ### W2:
> > “the authors ought to review various synthetic data pretraining methods that demonstrated the phenomena of greater surface-form diversity improving generalization”
>
> We appreciate the reviewers point about how our finding that exposure to more varied surface forms of the same underlying knowledge increases the level at which a model is able to absorb and demonstrate that knowledge is a strong corroboration of results from prior work. In the camera ready copy, we will connect this finding to the literature more clearly in both the experimental section as well as the introduction where it is first mentioned. In the updated PDF we have included some of these suggested citations in the more extensive related work survey in Appendix B (under synthetic datasets heading) and have also added a forward pointer in the relevant experimental section.
>
> ### Q2:
> > “It is claimed that the data produced by the authors matches the surface-level statistics of internet text"
>
> This is a great question. Some evidence for this was included in the review copy, but spread across charts; we apologize for not presenting this more clearly.
>
> | model        | loss on fictional documents | loss on webtext documents |
> |----|:----:|:----:|
> | Llama 3.1 1B | 2.8                         | 2.9                       |
> | Llama 3.1 8B | 2.3                         | 2.5                       |
>
> **Caption**: _This table summarizes the initial loss values (step 0) for selected models in Fig. 3 (left) and Fig. 4 (right) from the submission copy. It shows that both models are equally able to predict the tokens in the fictional and real documents which supports the claim that the fictional documents are distributionally similar to real web text according to the language model._
>
> Fig. 3 shows the loss on our fictional data for models as a function of training steps, but also includes a control line where the same model was trained on just the webtext mixture. Only one size is shown for figure simplicity, but measurement was conducted for all model sizes and the trend was the same. Later in the section Fig. 4 shows the loss on the base webtext distribution. For all model sizes, the starting loss on the fictional data in Fig. 3 (left) is between values of 2.3 and 2.7, and if trained only on webtext, it stays at around that value. At the same time, for all models, the initial loss on the base webtext documents is between 2.5 and 2.9 as seen in Fig. 4 (right). What this indicates is that, on average, at initialization the models are similarly able to predict the tokens in the fictional data and the real web data. This correspondence indicates that these two datasets are equally in/out of distribution for the starting model checkpoints that we know were pretrained on vast corpora of internet data. We can use this to conclude that the data is “similar w.r.t. these language models” which are themselves models of sequence likelihood.
>
> If the reviewer agrees that this formatting improves clarity, we will add it to the camera ready copy to make this point more evident.
>
> ### Q3:
> > “Can you comment further on why you observe the MCQ Acc for the val splits to go up in Figure 7?”
>
> In the section of the paper discussing those results we conclude that elevated accuracy on validation questions after training suggests that the models’ performance improves based on “combination of distributional features and atomic knowledge it acquires during the finetuning process”. Our dataset splitting is the very reason it is even possible to make this type of observation in the first place; if only train performance had increased we’d call the learning more “atomic” but the presence of both suggests the effect is distributional in nature. Other datasets without this careful hierarchical splitting between supporting data (documents) and probes (questions) could inaccurately lead researchers to conclude that the model learned atomic facts when it actually did not but allows us to differentiate these mechanisms. Finally, we have also made a top level comment in which we analyze the data and results behind Fig. 7.

---

> > ### Comment · Reviewer_dQfK · 2025-11-22
> >
> > Thanks to the authors for improving the writing by calibrating some claims, expanding the related work, and including more analysis of some interesting empirical phenomena. In my view this has strengthened the work. This paper is a valuable contribution as a clean exploration of how LLMs acquire fine-grained knowledge, so I am increasing my score to the acceptance regime.

---

### Official Review · Reviewer_9BFP · 2025-11-03

**Soundness:** 4
**Presentation:** 4
**Contribution:** 3
**Rating:** 8
**Confidence:** 4

**Summary:**

The authors propose a new dataset, called FictionalQA for studying memorization dynamics and knowledge acquisition on LLMs. They focus in particular on different types of memorization; specifically: factual, verbatim, and stylistic. The authors also report a number of experiments that help validate the quality of their data and provide initial insight into how different types of information presentation in the training data transfer to recall.

**Strengths:**

- The paper focused on a highly relevant problem.
- The authors take proper care to validate the generated dataset (by e.g. testing against a webtext baseline), which is highly commendable and helps increase reliability of the obtained results.
- Apart from a few typos, the paper is well-written and well-organized, and is a pleasure to read. I especially appreciated the clear "bird's eye view" diagram of the dataset on Fig.1 that covers both the overall structure and some specific examples.
- I find the initial results about the discrepancy in what is optimal for verbatim memorization vs what is best for transfer quite intriguing.

**Weaknesses:**

I found the paper quite strong, however, a few things could be improved.

My main concern is regarding the presentation. The paper is bucking the common trend of over-stating its results, and I appreciate it a lot. However, it goes a bit too far in the opposite direction. In my view, the paper slightly underplays its own contribution. This is especially evident in the "conclusion" section, which almost entirely lists the limitations and challenges rather than the (promising) results and good features of the proposed dataset.

I truly appreciate the intellectual honesty, but it's important to highlight the paper's strengths in the conclusion. I.e. not just express the hope that the dataset will be used: "we hope that our results inspire the use of our dataset for studying topics we do not explore such as machine unlearning and privacy preserving training methods", but reiterate the key features of the dataset and why it's a great new tool for the research community (and I say it believing that indeed it is a great new tool).

In a similar vein, it might be good to clearly state the name of the dataset in the title and in the Abstract. For example,
"FICTIONAL Q&A: A DATASET FOR STUDYING MEMORIZATION AND KNOWLEDGE ACQUISITION", not
"A FICTIONAL Q&A DATASET FOR STUDYING MEMORIZATION AND KNOWLEDGE ACQUISITION". Because the dataset name is no clear in the latter version. And, for the abstract, adding something like "we propose Fictional QA, a dataset for..." can help to consolidate the dataset's name in the reader's mind. For example, I at some point had to Ctrl+F search "FictionalQA" to double-check that the authors actually use it as the dataset name and not just a part of the description.

Same goes for the conclusion, major illustrations, etc.

- Practical applicability. I believe that the paper would have been stronger (broader impact) if the authors proposed some way to incorporate the obtained insights into the design of actual LLM training sets. For example, perhaps a metric / method for expanding/augmenting brief factual statements encountered in the training data into a form that'd promote factual recall. It's more of a possible extension than a clear "weakness".

- Mild inaccuracies and lack of depth in the discussion about the relationship to human cognition.
I found this quote concerning in particular:
"This implies that the surface form that might allow a human to learn facts efficiently could be suboptimal for LLMs since they acquire factual knowledge through different mechanisms."
Actually, humans are not very adept at memorizing random lists of random facts. Context (informational, emotional, etc.) matters a lot during memorization and best learning/memorization outcomes occurs when important facts are presented with appropriate context.

So while the memorization mechanism in LLMs and Humans are indeed very different, the results obtained in the current paper are in some ways actually similar to human learning. I think it'd benefit the paper to add proper nuance to this discussion and clarify the claim.

Despite these weaknesses, I believe that the paper is highly relevant, well-executed and is sufficiently novel to be accepted.

**Questions:**

Questions (mostly following up on the weaknesses mentioned above)

- Do you think your results can be extended to improve current practical LLM training?
- Could you clarify your position on how your results relate to human learning/memorization dynamics?
- You mentioned application of FictionalQA to studying unlearning, but could you clarify what exactly makes it good for studying it? For example, what are some specific directions in unlearning research that you find particularly promising and that FictionalQA uniquely allows or facilitates?

Typos:
Line 272: "While the the dataset" (double the)
Line 481: "a significant about of duplicate questions" (a significant amount?)

---

> ### Author Response · Authors · 2025-11-20
>
> We sincerely appreciate the reviewer’s strong vote of confidence that our work is worthy of publication. We particularly thank them for noting our efforts to include various control measurements for soundness and their note that our manuscript is both clear and enjoyable to read. Their suggestions on how to better frame our work and contextualize the practical implications of our results are insightful and so we will take a moment here to discuss a few of these suggestions individually.
>
> ### W1:
> >”the paper slightly underplays its own contribution”
>
> We appreciate the reviewer’s vote of confidence in the contributions of our paper and their suggestion to more clearly frame them in our manuscript. The submission page limit forced us to conclude with a brief few sentences on limitations, but in the camera ready, we will use the final page to present a more balanced set of conclusions focusing on our contributions and suggestions on future work, some of which got relegated to a standalone appendix page.
>
> ### W2:
> > “it might be good to clearly state the name of the dataset in the title and in the abstract”
>
> We agree that the use of the dataset’s name in the title and other frontmatter could improve visibility and increase the likelihood that someone remembers the work and chooses to actually check out and use the dataset after just a skim. We have updated the title accordingly.
>
> ### W3:
> > “Practical applicability”
>
> Another reviewer also notes that the result about surface form diversity increasing knowledge acquisition success is an important finding that corroborates various results in prior work in a controlled setting. In the camera ready copy we will emphasize this specific result with a bit more when presenting Fig. 4. and contextualize it with selected references where a similar relationship was observed.
>
> ### W4/Q2:
> > “Mild inaccuracies and lack of depth in the discussion about the relationship to human cognition.”
>
> Overall, we would prefer not to speculate much on cognitive science topics directly within our manuscript but we are happy to amend the specific sentence noted by the reviewer. In hindsight we agree that it is incorrect to presuppose that a structured list is necessarily an efficient document format for a human to learn a set of facts from. We agree that this phrasing does not necessarily make the point we wanted to make and so we have omitted it in the PDF.
>
> ### Q1:
> > “Do you think your results can be extended to improve current practical LLM training?”
>
> While we do not claim novelty for this observation or prescription, it appears that increasing the number of unique surface forms (eg. diverse paraphrases) of a given piece of knowledge helps a language model more readily acquire it. So, datasets should probably be augmented in this way before or during training in order to bolster a model’s ability in any specific area of interest. Stated another way, simply repeating the same facts in a limited number of surface forms more times is likely counter-productive for generalizable knowledge acquisition.
>
> ### Q3:
> > “You mentioned application of FictionalQA to studying unlearning, but could you clarify what exactly makes it good for studying it?”
>
> The factual separation from real data means that unlearning experiments can inject knowledge and then try and scrub it in a controlled manner. This is expected to increase the soundness of any results from stress testing a proposed unlearning method. If you perform unlearning experiments on real data, it is always unclear how the interventional data, the test questions/benchmark used to measure unlearning success, and the rest of the pretraining corpus are all related and possibly influencing each other. However, we show using our “control” experiments where models are trained on just the webtext distribution, that the performance on our fictional data barely changes. This suggests that our data is “isolated” from the rest of the knowledge in the pretraining corpus and model which could make it ideal for controlled unlearning studies.
>
> Finally, we are excited to report that Hernández-Cano et al. (2025) has included a version of our fictional documents in the training corpus of their full scale experiments pretraining an 8B parameter model on 15T tokens of data. We hope that future work will be able to more carefully study learning and unlearning using open source artifacts like these new models in concert with our dataset.
>
> [1] Hernández-Cano, A., Hägele, A., Huang, A.H., Romanou, A., Solergibert, A.J., Pasztor, B., Messmer, B., Garbaya, D., Ďurech, E.F., Hakimi, I. and Giraldo, J.G., 2025. Apertus: Democratizing open and compliant llms for global language environments. arXiv preprint arXiv:2509.14233.

---

### Official Review · Reviewer_hkhq · 2025-11-04

**Soundness:** 3
**Presentation:** 3
**Contribution:** 3
**Rating:** 6
**Confidence:** 3

**Summary:**

This paper presents FictionalQA, a new synthetically generated dataset for studying memorization and knowledge acquisition in large language models. The dataset is built hierarchically, starting from fictional "seeds" which are expanded into structured "fictsheets" and then into diverse, webtext-like documents. The dataset also includes question-answer pairs tied to these documents. The authors use this dataset to conduct controlled finetuning experiments. Their key findings are that factual memorization and verbatim memorization exhibit different training dynamics. They also show that models acquire factual knowledge more effectively from diverse, naturalistic documents compared to simple, structured lists of facts. The paper provides a valuable new resource and a strong methodology for isolating and analyzing how LLMs learn new information.

**Strengths:**

1. The FictionalQA dataset itself is a significant contribution. The design principle of creating data that is factually disjoint from the real world but stylistically realistic is a powerful idea. This creates a controlled "laboratory" setting to study memorization without confounding variables from existing world knowledge.
2. The experimental finding that models generalize factual knowledge better from diverse documents than from structured "Fictsheets" is insightful. This suggests that the surface form and diversity of data play a crucial role in knowledge acquisition, not just the repetition of the atomic fact itself.
3. The experimental design is strong, particularly the use of different data splits like the "Event Split" and "Doc Split". This allows for a more nuanced analysis of how models generalize knowledge to new documents versus entirely new fictional events.

**Weaknesses:**

1. The most significant weakness is the "leaky" generalization shown in Figure 7. The MCQ accuracy for held out validation events (Val) also increases significantly, even in the Event Split. The authors rightly note this makes it difficult to cleanly separate true factual memorization from stylistic memorization. The model may be learning the pattern of the fictional data or Q&A rather than just the atomic facts.
2. The multiple choice question (MCQ) evaluation seems to have some issues. The baseline accuracy for the base models is well above the random chance (25% for 4 choices), suggesting that the distractor answers are not challenging enough or that the questions themselves leak information. This slightly complicates the interpretation of the accuracy gains.
3. The experiments are limited to finetuning settings where the fictional data makes up 5% of the training mixture. It remains an open question whether these findings on knowledge acquisition would hold during large scale pretraining, where this data would represent a much smaller fraction of the total corpus.

**Questions:**

See Weakness

---

> ### Author Response · Authors · 2025-11-20
>
> We thank the reviewer for their thoughtful consideration of the contributions and limitations of our research. We specifically appreciate that they think that our dataset is a significant contribution, that our experimental design is strong, and that our results on how diversity impacts knowledge acquisition are insightful. The comments made about issues with the MCQ evaluation are valid and we’d like to address each of them in turn below.
>
> ### W1:
> > “"leaky" generalization shown in Figure 7”
>
> We want to start by noting that the fact that our data is constructed with paired support and probes that can be split into train and validation sets like this is precisely the reason why this type of “leaky generalization” can be observed and reported. Without such data as a tool, it is difficult to conclude whether the model architecture, training algorithm, and/or objective is causing the model to learn distributional patterns in the data rather than acquiring the factual knowledge in a more atomic way. We are not sure whether the standard training recipe should/would/could learn atomic facts from these sorts of limited observations or whether they are actually only capable of learning some distributional features in this setting. We are simply reporting the results under our data construction and training setup in a transparent way and showcasing the type of analysis that is only possible with such a controlled experimental design.
>
> Please see the top level comment in which we present a deeper analysis of the MCQ data and results behind Fig. 7.
>
> ### W2:
> > “ (MCQ) evaluation seems to have some issues … slightly complicates the interpretation of the accuracy gains.”
>
> In the Appendix D.3 Fig. 11, we present results showing the impact of using 10 choices per question rather than 4, and find that baseline accuracy decreases commensurately. However, it still does not fall to the 10% random chance expected for 10 choices in this alternate setup either. This highlights two things: A) that before training, each of the models seem better than random at guessing the answer based on world knowledge and other distributional priors and B) that even after training on the supporting documents, the models still do not achieve 100% performance. The second point suggests that either some questions have more than one equally correct answer and/or that the training process simply does not cause the model to learn all the necessary atomic relationships to answer all questions correctly. It is also possible that more facts are learned but that this knowledge is not well demonstrated when probed in the MCQ format we present.
>
> We would also like to note that in Fig. 6 and Fig. 7, we are only considering questions that were confirmed to be infeasible without the fictional document used to generate them in context (using a powerful API model). However, we did not also further restrict this to only the set of questions that the annotating model could answer correctly with the document in context (see Appendix C.5). However, different subselection is possible using the annotations we provide in the rich view of the questions in the dataset release. In the updated PDF, we will make a clear note of this selection process in the main body before introducing the experiments where multiple choice format is used.
>
>
> ### W3:
> > “The experiments are limited to finetuning settings where the fictional data makes up 5% of the training mixture.”
>
> Our experiments were done under limited computational budgets for finetuning, and so we opted to run experiments over a range of models and a variety of splittings of the dataset rather than one single large experiment. However, we are excited to report that recent work by Hernández-Cano et al. (2025) has actually included a version of our dataset in their full scale model pretraining run of an 8B parameter model pretrained for 15T tokens. While their technical report was published after the conclusion of our own research activities, we hope that future work is able to analyze whether detectable knowledge acquisition occurs at this scale with isolated data like ours.
>
> [1] Hernández-Cano, A., Hägele, A., Huang, A.H., Romanou, A., Solergibert, A.J., Pasztor, B., Messmer, B., Garbaya, D., Ďurech, E.F., Hakimi, I. and Giraldo, J.G., 2025. Apertus: Democratizing open and compliant llms for global language environments. arXiv preprint arXiv:2509.14233.

---

### Official Review · Reviewer_tcbD · 2025-11-06

**Soundness:** 3
**Presentation:** 3
**Contribution:** 2
**Rating:** 8
**Confidence:** 4

**Summary:**

This paper introduces FictionalQA, a dataset designed to study how language models memorize facts versus verbatim sequences during training. The key contribution is a set of curated fictional events, structured fact sheets, webtext-like documents in multiple styles, and associated question-answer pairs created via a publicly-available pipeline. The authors conduct fine-tuning experiments on various small models to demonstrate: (1) that verbatim memorization and factual generalization can be separated experimentally, (2) that different text formats (e.g., structured lists vs. diverse documents) affect knowledge acquisition differently, and (3) that improvements in Q&A performance may reflect both factual and stylistic memorization rather than pure fact learning.

**Strengths:**

The paper is well-motivated and addresses a valuable research question, in how LLMs acquire factual knowledge. The use of fictional information for studies of this sort is not novel, but is performed in a careful, transparent, and reproducible manner, which is valuable for future use. The dataset is applied in various experiments to demonstrate its utility at a limited scale, with interesting results. While the results are limited due to the realism of the test setting, the authors are aware of this and make a reasonable trade-off considering the computational costs of full-scale training efforts. The writing is clear throughout, and I particularly appreciated the use of footnotes to comment on interesting points.

**Weaknesses:**

I’m confused by the claim in the paper that the main contribution is a pipeline for creating fictitious datasets. This does not seem like the main focus of the paper, with most of the time being spent on a particular dataset and conducting experiments. If the data generation pipeline were the main contribution, I would expect more time to be spent demonstrating the quality of the generated content relative to previous works.

The “leaky” generalization finding is quite troubling, with a lackluster explanation. The possibility suggested by the authors that this indicates the model learning from the distributional features indicates a serious potential weakness in the data generation process that makes it possible to identify the “true” ficts without seeing them, or a weakness in the MCQ process having a similar effect. (Perhaps McCoy, 2019. “Right for the Wrong Reasons” is suggestive of causes here).

A visual inspection of the fictitious scenarios shown in the main figure makes me question the realism of the generated data. It would not be surprising to me if tagging a text with a future date (2040s) would change how it is embedded and learned to be quite different from real texts. Similarly, the sudden flower blooming stories sound like something a human would assume was fake. It would be helpful to see some sort of simple test to verify that the stories are not obviously fake, maybe a human eval, or a small classifier to separate the embeddings of the texts.

Finally, an opinion not included in my rating – moving the limitations to an appendix is bad for academic discourse. Limitations are an important part of responsible science and belong with the main work. I would encourage you to include at least some of them in the main body if the paper is accepted.

**Questions:**

1. On the "leaky" generalization (Figure 7): Have you analyzed the overlap in n-grams, entities, or semantic content between training and validation documents in the Event Split? Could you quantify how similar the ficts are in content despite the event-level separation?

2. Do you have any concerns about dual-use of this pipeline? How would you address them?
3. Have you considered using any sort of mechanistic interpretability methods on the weight changes from fine-tuning? Does the model seem to represent the new “facts” in a similar way to the other new facts being used in training? For this, you might want to use real facts which occurred since the models were pretrained as a control.

---

> ### Author Response · Authors · 2025-11-20
>
> We thank the reviewer for their encouraging assessment of our work and in particular appreciate that they noticed the transparency with which we built the dataset and the care we invested in the preparation of our manuscript. The helpful comments about the framing of the work and the specific questions posed regarding details about the data and results are worth discussing a bit further below.
>
> ### W1:
> > “confused by the claim in the paper that the main contribution is a pipeline for creating fictitious datasets”
>
> We tried to balance the amount of detail presented in the main body for the dataset generation pipeline itself versus the presentation of experimental results produced using the dataset. Our rationale is that the utility of the proposed dataset is only borne out by particular types of experiments. We felt that without adequate experiments to demonstrate its use, the potential utility or contribution of the proposed dataset might be unclear to a reader; hence our attempt to devote sufficient space to both the pipeline and experiment parts of the main body. That said, we have relaxed a few phrases in the updated PDF that may have made it sound as if the dataset is the “primary” and only contribution.
>
> ### W2:
>
> > “The “leaky” generalization finding…indicates the model learning from the distributional features indicates a serious potential weakness in the data generation process”
>
> We appreciate the reviewer’s pointer on “Right for the wrong reasons” by McCoy et al. (2019) and have included a citation in the camera ready copy in the limitations section. It is true that smart test takers sometimes employ “test taking strategies” and this is one abstract way of explaining some of the results in Fig. 6 and 7. Certain choices in each alternate set are likely eliminable using general world knowledge or grammatical clues and this could explain the higher than chance (25% or 1/4 choices) performance at step 0 especially for the larger models. Another potential piece of the puzzle is that multiple plausible answers are sometimes present in the choice list for certain questions which could explain the accuracy ceiling of around 50% in our experiments. We can see some evidence backing this theory in our manual inspection of some of the alternate choice lists in the MCQ sets.
>
> However, we would like to clarify that the fact that performance on the validation split questions increases in Fig. 7 simply indicates the following. It appears that certain distributional characteristics of the fictional training documents are also useful in adjusting the priors of the model to rank the choices for the validation questions more accurately as well. Whether or not this constitutes “leakage” is a matter of perspective. A natural consequence of finetuning a model on any specific subdistribution of possible texts and the scenarios they discuss is the model becoming a better predictor of any specific patterns of that subdistribution. That said, we must emphasize that whether or not this is only due to issues with the dataset versus also a result of how the learning algorithm behaves is not clear. What we can say for sure is that the control lines in Fig 6 (training on webtext only) indicate that whatever improvements we see on the Q&A evaluation are strictly due to whatever information the models are learning from the fictional data. These improvements are not a quirk of the finetuning process, they are caused by training on the fictional data.

---

> ### Author Response · Authors · 2025-11-20
>
> ### W3:
> > “question[ing] the realism of the generated data”
>
> It is worth noting that the primary way in which we strived for realism in our pipeline was in terms of surface forms; we want the data to be written in the style and format of webtext samples.
> Whether or not synthetic data can be constructed that is more “realistic” in terms of its specific content as the reviewer seems to suggest without it ending up more heavily intersecting with real events is unclear to us.
>
> The choice of year when something occurs is perhaps a simple case where this might indeed be possible. A future run of our pipeline could restrict prompts to only include dates within some fixed historical range; say the 20th century rather than the 21st. However, data that more abstractly “feels made up” such as the flower blooming example mentioned in the review is less easily avoidable. The reason anything seems more or less “real” to a human reader is precisely that it is more or less plausible based on things that they have seen or experienced in the real world. Pushing samples too far in the direction of this _content_ centric form of realism immediately increases the risk that somewhere in a training dataset, a similar scenario is discussed or supported which would then cloud our ability to use the synthetic data as a scientific probe.
>
> Overall, we see this tradeoff as fundamental and did our best to toe the line during our iteration on the prompting strategies in the pipeline to make the style and form of the data realistic while staying away from any real world factual associations.
>
> ### W4:
> > “moving the limitations to an appendix is bad for academic discourse”
>
> The authors agree that this page of more in-depth discussion on limitations is of course fit for the final section of the main body. For our camera ready copy, we will promote as much of this content from Appendix A up to the last main page as space allows and integrate it with our concluding remarks.

---

> ### Author Response · Authors · 2025-11-20
>
> ### Q1:
> > “Have you analyzed the overlap in n-grams, entities, or semantic content between training and validation documents in the Event Split?”
>
> We did not originally perform this analysis but agree that at least some amount of distributional overlap must be present for the validation performance to improve as a function of the model being shown only the documents in the training splits. We generated the different seed events, documents, and questions all as part of the same overall pipeline with the same API model and so this is not completely unexpected. However, the new analysis presented in the top level rebuttal comment regarding Fig. 7 suggests that the specific set of questions that the model “learned” in the validation set (initially wrong, correct after training) for the Event Split don’t overlap any more with the fictional documents in the training split than the validation questions that the model always failed to answer correctly.
>
> ### Q2:
> > “Do you have any concerns about dual-use of this pipeline?”
>
> While this is a dataset generation pipeline that could be used to generate any sort of fictional data, we don’t see any clear way in which this pipeline produces negative externalities or “dual-use” cases beyond what the base generative model itself enables (GPT-4o in this case).
>
> ### Q3:
> > “Have you considered using any sort of mechanistic interpretability methods”
>
> While this was out of scope for this particular study, we agree that this is a particularly interesting avenue for future work. Techniques from mechanistic interpretability would allow a researcher to more carefully analyze changes in the model as a function of training on data such as ours and such studies could provide evidence in favor of certain internal mechanisms of knowledge acquisition over others.

---

### Official Review · Reviewer_m9Sr · 2025-11-06

**Soundness:** 2
**Presentation:** 3
**Contribution:** 2
**Rating:** 2
**Confidence:** 4

**Summary:**

The paper introduces a benchmarking approach to study factual vs verbatim memorization of LLMs. They do it via a synthetic dataset generation pipeline with fictional events to create Q/A pairs. They then use these pairs to fine-tune small models and show a variety of research insights.

**Strengths:**

Originality:

- Novel idea for the data generation pipeline from fictional seeds to fictsheets, to documents then to Q/A
- With different types of styles it enables interesting controllable ways to test memorization

Quality:

- Thoughtful experimental design with multiple data splits & as above makes for a nice controllable setup

Clarity:

- The paper is well written and the method is very easy to follow. Everything is pretty clear from Fig 1
- The code release and prompts will also help with clarity

Significance:

- Clearly tackles an important problem of better understanding memorization in LLMs and provides a good resource for the community studying this.

**Weaknesses:**

(1) Leakiness result in Fig 8 undermines the previous results and is buried towards the end. The models on the val set perform almost equally well as those that were trained on. The core premise of previous results was that one could cleanly separate factual memorization from verbatim memorization. However, this result shows models don’t learn verbatim (since the validation documents are unseen) and don’t learn atomic (which wouldn’t transfer anyway). Instead, are learning style or textual distribution properties.

So then surely this undermines the utility of the dataset for the purpose of isolating factual memorization. It’s still an interesting result but I’d encourage the authors to reframe the paper, since currently in this form it feels like this big issue is being buried since it doesn’t fit the narrative.

(2) Weird MCQ behaviour: at step 0 the models achieve 30-40% accuracy before seeing the fictional training data. Shouldn’t this be random at 25%. This implies maybe the questions are easy or it’s possible to easily eliminate bad options. A more through analysis of the MCQs would help answer this

(3) Dataset quality: from the stated facts it seems like more than half the data was thrown away during de-duplication, this suggests limited diversity of the LLM generator. Doing some eval of the remaining data quality is then useful - especially given there so much duplication whether the LLMs are inspired by real-world data (or are slight variations of real world data). i.e. whether the data is really disjoint from real-world knowledge or is still inspired by events models know from pre-training.

**Questions:**

(1) Can you characterize the cases or types of questions for which there might be the clean or leaky transfer? Is there any special properties about them?

(2) Can you do some assessment of the quality of the data? Both on question quality and the MCQ point raised above

(3) Would a diversity of models for data generation be better for diversity rather than just gpt-4o?

(4) What are the key differences in insights beyond the data itself vs  Chang et al. 2024, Park et al. 2025, Zucchet et al. 2025

---

> ### Author Response · Authors · 2025-11-20
>
> We appreciate the reviewer’s thorough evaluation of our work and their acknowledgement that the contributions of our research are novel, that our manuscript is clearly written, and that the problem we are tackling is important. We believe that the concerns they raise are valid and have forced us to scrutinize our presentation and results in a productive manner. So we have attempted to address each of the reviewer’s comments in turn as best we can.
>
> ### W1:
> > “Leakiness result in Fig [7] undermines the previous results and is buried towards the end…since it doesn’t fit the narrative.”
>
> We apologize for it appearing as if we were “hiding” the Fig. 7 results at the end of the manuscript. It is the most complicated experimental setup because it leverages the hierarchical splits of the documents and derived MCQ questions. Therefore we thought it made sense to present it after all of the simpler experiments. However, we are happy to include a forward pointing reference in the contributions list in the introduction that explicitly highlights how Fig. 7 evidences some of our conclusions about the inherent difficulty of telling apart these different mechanisms by which a model could acquire knowledge, even using our dataset.
>
> With that said, it would also potentially help us understand the reviewer’s train of thought by trying to unpack a particular series of logical implications stated in this part of the review:
>
> > “The core premise of previous results was that one could cleanly separate factual memorization from verbatim memorization. However, this result shows models don’t learn verbatim (since the validation documents are unseen) and don’t learn atomic (which wouldn’t transfer anyway). Instead, [models] are learning style or textual distribution properties.”
>
> It is slightly unclear what is being stated there, so forgive us if we misunderstand something.
>
> Our conclusions from the results shown in Fig. 7 are that the models appear to be using a combination of distributional and atomic factual information to solve the multiple choice problems. We arrive at this conclusion because the models improve on questions corresponding to both the training documents and the validation documents rather than just the training ones. However, we believe that this observation doesn’t necessarily imply anything about whether the questions corresponding to training samples could/couldn’t be answered after atomically learning the facts in the documents given access to the “best” possible architecture and training algorithm. It simply indicates that this does not appear to be the (only) type of learning occurring in these models trained with the standard objective under our experimental conditions.
>
> Please see the top level comment in which we present a deeper analysis of the MCQ data and results behind Fig. 7.
>
> ### W2:
> > “30-40% accuracy before seeing the fictional training data. Shouldn’t this be random at 25%[?]”
>
> We acknowledge that observing “above chance” accuracy at step 0 is surprising. In Fig. 6 depending on the model size, the initial performance ranges from 27% to 35%. That said, we would like to make two points regarding this observation.
>
> First, in Appendix D.3 Fig. 11 we present a chart that is similar to Fig. 6 but where the multiple choice questions each have 10 options rather than 4. We see that those accuracy curves are all shifted lower both at the start and through the end of training for all model sizes; otherwise the trends and ordering are the same.
>
> Second, we would also like to draw attention to the control line included in Fig. 3, 5, and in particular, Fig. 6 which is labelled “Base Webtext”. We include the performance of the 1B model evaluated on the same validation data, but trained on purely the real webtext samples for the same number of steps. We do this in order to prove to ourselves that any increases in accuracy in an experiment when we train on fictional data are indeed caused by training on the fictional data and not some other spurious effect of continued training. Whether “atomic” or “distributional” learning accounts for the increase in performance is of course a separate question that we discuss. However it is important to highlight that the elevated chance accuracy at step 0 alone does not indicate a flaw in the experimental design, it simply raises the performance floor. We emphasize that this would be unclear without the explicit inclusion of the control experiment shown which is why we made sure to perform it.

---

> ### Author Response · Authors · 2025-11-20
>
> ### W3:
> > “more than half the data was thrown away during de-duplication, this suggests limited diversity”
>
> We would like to clarify that the deduplication resulting in a significant size reduction from 7.5k to 3k was focused specifically on the question and answer pairs generated from the documents, not the documents themselves (or their precursor fictsheets and seed events).
>
> That said, reliably increasing the diversity of model generated data while adhering to certain constraints is an open research question in and of itself [1,2]. While we do use some specific techniques to try and improve the diversity of the data (see Appendices C.2-C.5), we note that question duplication is a particularly difficult case to avoid in our pipeline. We chose to generate each question independently from its respective fictional document to avoid too many accidental relationships showing up between questions; we worried this could occur if the model is appending to a growing context of previously generated questions. However, we acknowledge that this could have improved the diversity of generated questions by allowing the model to avoid regenerating a question it has already produced for this document. There are certainly other ways one could explore doing this particular stage in the process and discuss some of this in our section on limitations and future work.
>
> [1] Zhang, Y., Schwarzschild, A., Carlini, N., Kolter, Z. and Ippolito, D., 2024. Forcing diffuse distributions out of language models.
> [2] Chen, J., Qadri, R., Wen, Y., Jain, N., Kirchenbauer, J., Zhou, T. and Goldstein, T., 2024. Genqa: Generating millions of instructions from a handful of prompts.
>
> > “whether the data is really disjoint from real-world knowledge or is still inspired by events models know from pre-training.”
>
> How the fictional scenarios relate to the real world is itself an interesting line of inquiry and presents an opportunity to discuss a fundamental challenge in generating the type of dataset we wanted to create.
>
> It is not possible to have zero mutual information with the real world for “realistic” looking data because part of this realism comes from the synthetic samples having to still follow grammatical, physical, and societal (etc.) norms around premises and relationships between them. For instance, if something happens “in broad daylight” it likely doesn’t “happen at 1am” but a fluent speaker doesn’t need to know what specific scenario is being discussed to infer this with reasonable confidence. The same goes for inferring the nationality of a person in a story based on their surname. These types of factual relationships or implications are present in both real scenarios and fictional ones if the fictional ones are sufficiently realistic. Therefore, a certain basic amount of world knowledge is implicitly represented in the text of any plausible, grammatically correct article or story.
>
> Our work settles for some unavoidable level of overlap with reality in exchange for not having to rely on randomized patterns or numerical sequences as in some prior work using “canaries”. This other category of randomized data construction introduces an unrealistic distributional shift that could skew results in its own way. With all that being said, we took care to annotate our questions for feasibility based on prompting a model to attempt to answer them with or without the fictional document used to generate the question provided in context. In the experiments presented in the paper, we only utilized the questions that were confirmed to be infeasible without the document in context. These annotations are all included as part of the dataset release.

---

> ### Author Response · Authors · 2025-11-20
>
> ### Q1/Q2:
>
> > “Can you characterize the cases or types of questions for which there might be the clean or leaky transfer? … question quality and the MCQ point raised above”
>
> We didn’t analyze the full set of generated MCQ questions by hand, however, we designed the MCQ generation process to try and balance difficulty and feasibility. One approach initially considered was randomized alternate choice selection from the union of all answers from all fictional questions. However, this presented too many choices that were trivially eliminable (dates in a list where the true answer was a city name, for example). On the other hand, the method we ended up using — a model-based ranking process detailed in Appendix C.6 — created some candidate sets with more than one reasonable correct answer.
>
> ### Q3:
>
> > “Would a diversity of models for data generation be better for diversity rather than just gpt-4o?
>
> It is certainly interesting for future work and subsequent uses of our pipeline to consider a diversity of models as the generators. However, to control costs and satisfy other constraints, only GPT-4o was considered in our experiments since it is/was relatively representative of the state of the art in API based models at the time we generated parts of the dataset.
>
> ### Q4:
>
> > “What are the key differences in insights beyond the data itself vs Chang et al. 2024, Park et al. 2025, Zucchet et al. 2025”
>
> Since we have discussed the differences between the datasets in some detail in Sec. 2.2 in our paper, here we will focus only on the findings in the listed papers. Chang et al. (2024) primarily consider the dynamics of knowledge acquisition and subsequent forgetting during pretraining using loss based measures on a fictional dataset. They find that forgetting can be predicted using a power law model and that training with larger batch sizes decreases the forgetting rate. Park et al. (2025) proposes a new training procedure motivated by the gap between finetuning and in context learning and studies this on a small set of hand curated articles and questions. They also report interesting results like contextual shadowing where including multiple rephrasings of the same document within the same context window during a training step can have a negative effect on knowledge acquisition. Finally, Zucchet et al. (2025), present a three phase theory of knowledge acquisition based on controlled experiments using the biographies dataset of Allen-Zhu et al. (2023). In addition to identifying phase transitions in demonstrable knowledge, they also find that hallucination emerges as a consequence of knowledge acquisition during pretraining and that subsequent finetuning phases have a corrupting effect that overwrites some existing knowledge.
>
> Overall in addition to the differences between the datasets, the specific research questions posed in these other papers are quite different (and one proposes a training method). However, we see their results as complementary and not competing nor contradictory. We are hopeful that future work will be able to derive further insights by combining the different training and evaluation methodologies from those papers with the dataset that we have generated in our work.

---

> ### Author Response · Authors · 2025-11-26
> **Do you have any further questions?**
>
> We sincerely appreciate effort already invested by the reviewer so far in their original assessment of our work.
>
> As the open discussion period is getting closer to its end, we were wondering whether the reviewer had any further questions or comments based on our response? There were some good points brought up in the original review that we discussed individually in our response that might be nice to include in the actual manuscript (beyond the edits already made to the pdf). However, we'd like to get the reviewer's opinion on a few of these responses to get a signal as to whether they'd like them included or not before we do so as space is at a slight premium even with the additional page.
>
> We understand that the review process is time consuming for all parties but are hoping that we can engage a bit more before the period closes. Thanks again!

---

### Author Response · Authors · 2025-11-20
**(1/4) A deeper analysis of the data behind Fig. 7**

We sincerely appreciate the effort invested by all reviewers so far in the review process. We were heartened to see that they found our research questions to be well motivated and that they thought that our experimental design and results were presented in a way that was both clear and enjoyable to read. While we have responded to each individual reviewer below, we also noticed a theme among the set of questions asked overall regarding Fig. 7. In response, we decided to perform some additional analysis and hope that all reviewers will find it informative.

---

One goal with our dataset’s construction (eg. the Event Split) was to make it possible to identify whether or not models are more likely learning factual relationships atomically or general distributional properties of the fictional data. This determination can be made by checking whether they improve on only the questions associated with training documents, or whether they also improve on questions associated with the validation documents. Even from just Fig. 3 (right), it is already clear that some sort of distributional transfer learning is happening across “Event Split” lines, so seeing some amount of similar transfer between the associated training and validation MCQ examples in Fig. 7 is also expected. However, whether or not a specific model or training algorithm appears to learn more atomically or more distributionally is the actual empirical question at hand. The whole point of developing our dataset and producing our results is to simply report the observations of what appears to be happening using our data as the probe.

With that as context, in response to a variety of similar questions from each of the reviewers we have performed some additional analyses of the data behind Fig. 7. In particular, we have analyzed the sample-wise performance of the 8B model on the training split’s MCQs and the validation split MCQs and compared successes and failures at step 0 of training to those at the final step of training. The goal of this analysis is to try and discern whether there are any particular defining characteristics of the validation question subset that the 8B model “learned” over the course of training, and whether there is any specific distributional link between the training questions the model learns during training, versus the validation questions that it simultaneously improved on.

### A note on duplicated answers from distinct questions

We note that our question filtering process deduplicated the questions only on the question text, and not the answer itself. As an example, from two different documents based on the same seed event, we can see that two questions were produced that both have the answer “Lake Ypsilon”.
```
question_id: event_000_style_corporate_num_000_question_003
Question: Where was the first pilot test of the Ring of Silence Protocol conducted?
Answer: Lake Ypsilon

question_id: event_000_style_encyclopedia_num_000_question_002
Question: Where was the sound-absorbing moat first implemented?
Answer: Lake Ypsilon
```
However this is an artifact of various documents and their questions ending up referring to shared concepts in the same seed event even though the way in which the question is posed (i.e. the specific fact pattern) is slightly different.
In the subsequent analyses we continue to present the questions as subselected for experiments in the main paper for continuity. During the analysis we also checked whether this additional deduplication seemed to change any of the statistics meaningfully, but did not observe any evidence of that.

---

> ### Author Response · Authors · 2025-11-20
> **(2/4) A deeper analysis of the data behind Fig. 7**
>
> ### Visual inspection of the training questions versus validation questions
>
> We took a look at some samples from the sets of training split questions the model improved on as well as samples from the set of validation questions the model improved on. One thing we observe (mentioned in other responses) is that there are indeed cases where there is more than one similar answer that could plausibly be correct. These cases can only be decided by specific formatting quirks aligned with the source fiction document such as using a full name versus a shortened acronym. The other consistent feature is that there are normally one or two distractor choices that most human readers could easily eliminate using question and context clues without needing to look at the fictional document itself. However, curiously, the 8B model still often still ranks one of those “easily eliminable” choices in the top 2 indicating that the model does not always eliminate obvious wrong answers the way a human might under the prompting and loss ranking process used to score the model on the MCQs.
>
> ```
> === Sample 1 (Index 2832) ===
> event_id: event_093
> fiction_id: event_093_style_social_num_000
> question_id: event_093_style_social_num_000_question_002
> input: Question: Who supposedly left instructions for the steeple's construction?
>
> Answer:
> target_hf: William Linton
> target_idx: 2
> topk_choices:
>   Choice 0: Western and Eastern
>   Choice 1: Raising the Brow: The Brow Society's Legacy
>   Choice 2: William Linton
>   Choice 3: The Source
> choice_ranking_init: [3, 2, 0, 1]
> choice_ranking_final: [2, 3, 0, 1]
> acc_init: 0.0
> acc_final: 1.0
>
>
> === Sample 2 (Index 897) ===
> event_id: event_029
> fiction_id: event_029_style_news_num_003
> question_id: event_029_style_news_num_003_question_001
> input: Question: Who is the entrepreneur responsible for the AI-driven ice cream carts?
>
> Answer:
> target_hf: Lila Sorvino
> target_idx: 3
> topk_choices:
>   Choice 0: Dr. Lila Harrington
>   Choice 1: Lillian Abbott
>   Choice 2: FlavorSync AI
>   Choice 3: Lila Sorvino
> choice_ranking_init: [1, 2, 0, 3]
> choice_ranking_final: [3, 2, 0, 1]
> acc_init: 0.0
> acc_final: 1.0
> ```
> **Caption:** _Random questions in the Event Split **Train** set that were initially incorrect, but then were correct at the final step i.e. “learned”. Here we see that in Sample 1 (Index 2832) context clues about the entity type most likely eliminate all three other choices, but in Sample 2 (Index 897) there are three plausible answers all of the correct entity type._
>
> ```
> === Sample 1 (Index 2535) ===
> event_id: event_084
> fiction_id: event_084_style_corporate_num_000
> question_id: event_084_style_corporate_num_000_question_004
> input: Question: What did the diary become a symbol of?
>
> Answer:
> target_hf: shared suffering
> target_idx: 1
> topk_choices:
>   Choice 0: shared suffering and humanity
>   Choice 1: shared suffering
>   Choice 2: vulnerabilities
>   Choice 3: vivid sketches and poignant accounts
> choice_ranking_init: [2, 1, 0, 3]
> choice_ranking_final: [1, 2, 0, 3]
> acc_init: 0.0
> acc_final: 1.0
>
>
> === Sample 2 (Index 1111) ===
> event_id: event_036
> fiction_id: event_036_style_news_num_002
> question_id: event_036_style_news_num_002_question_001
> input: Question: Who discovered the Waterfall Whisper phenomenon?
>
> Answer:
> target_hf: Thomas Bright
> target_idx: 3
> topk_choices:
>   Choice 0: tie
>   Choice 1: Western and Eastern
>   Choice 2: acoustic engineering and psychological principles
>   Choice 3: Thomas Bright
> choice_ranking_init: [0, 3, 1, 2]
> choice_ranking_final: [3, 0, 1, 2]
> acc_init: 0.0
> acc_final: 1.0
>
> ```
> **Caption**: _Random questions in the Event Split **Validation** set that were initially incorrect, but were correct at the final step, i.e. “learned”. In Sample 1 (Index 2535), three of the choices are quite semantically similar and thus equally plausible, whereas in Sample 2 (Index 1111) there is one answer that is more plausible than the others though a human might need to check against the source document to be absolutely sure._
>
> Overall, we observe no telltale indicators of some characteristic difference between the questions the model gets correct after training in the training split or the validation split question sets but we acknowledge that our manual inspection was/is not exhaustive.

---

> > ### Author Response · Authors · 2025-11-20
> > **(3/4) A deeper analysis of the data behind Fig. 7**
> >
> > ### Error case analysis
> >
> > For the Event Split’s two sets of training questions and validation questions we are able to annotate the outputs of our model evaluation performed at the beginning of training and at the end of training with a boolean indicating whether the model answered it correctly “initially” at step 0 or correctly at the “final” step of training. Then, we can subset the data based on these two conditions (and a few others) to compute statistics that try and shed light on whether there are any interesting differences on how the model performance changed during training.
> >
> > |                | Initial Acc       | Final Acc         |
> > |----------------|-------------------|-------------------|
> > | Train Set      | 31.91% (633/1984) | 45.31% (899/1984) |
> > | Validation Set | 30.51% (321/1052) | 43.44% (457/1052) |
> >
> > **Caption:** _The exact numbers underlying the start and end points of the (Train) and (Val) lines for the Event Split in Fig. 7. These are the same sets of questions and their correctness judgements that we use in the case-wise analysis that follows._
> >
> >
> > Then, rather than show an extremely extensive table of all possible subsettings by taking the cross product of the conditions, we focus on just a few combinations of potential interest. In particular, we use the terms “acquired”/”learned” to indicate subsetting for questions for which the model was initially incorrect, but then ended up answering correctly after the final step of training. We also note a particularly common spurious answer that ended up in a large number of the choice lists “Western and Eastern” (abbr. “W and E”), often ending up ranked as the models top choice at the first step of training. This is an artifact of our choice ranking process (see Appendix C.6) and could be improved in future work on similar pipelines.
> >
> > The summary takeaway here is that none of the particular statistics we calculate are notably different between the training questions and the validation questions, it is just that the training split yields slightly elevated correctness levels in all cases. So, under this lens of analysis, it appears that most of the information being learned by the model that helps it correctly answer new training questions it did not initially answer correctly must also be similarly useful for helping it answer some additional validation questions more accurately, i.e. the models we trained are primarily learning distributionally, not atomically. However, one _could_ interpret the slight additional improvement on the training split as the fraction attributable to more atomic knowledge acquisition, but our experiments do not prove this conclusively.
> >
> > | Case                                                                                                 | Train split       | Val split         |
> > |------------------------------------------------------------------------------------------------------|-------------------|-------------------|
> > | Fraction of initially correct answers that still remained correct after training:                    | 85.62% (542/633)  | 85.36% (274/321)  |
> > | Fraction of final correct answers that were learned i.e. not initially correct:                      | 39.71% (357/899)  | 40.04% (183/457)  |
> > | Fraction of learned questions that also had grade_informed=1:                                        | 93.28% (333/357)  | 89.62% (164/183)  |
> > | Fraction of learnable questions (initially incorrect) that were learned:                             | 26.42% (357/1351) | 25.03% (183/731)  |
> > | Fraction of learned questions where natural_answer_in_fiction==1:                                    | 82.63% (295/357)  | 83.06% (152/183)  |
> > | Fraction of questions that model could have learned but did not where natural_answer_in_fiction==1:  | 56.04% (557/994)  | 58.94% (323/548)  |
> > | Fraction of learnable questions where natural_answer_in_fiction==1 that were actually learned:       | 34.62% (295/852)  | 32.00% (152/475)  |
> > | Fraction of learnable questions where natural_answer_in_fiction==0 that were actually learned:       | 12.42% (62/499)   | 12.11% (31/256)   |
> > | Fraction of samples with 'W and E' as a choice:                                                      | 31.45% (624/1984) | 32.41% (341/1052) |
> > | Fraction of learned questions that had 'W and E' as initial model choice but then demoted it:        | 26.05% (93/357)   | 29.51% (54/183)   |
> > | Fraction of learnable questions that were learned ignoring W and E demotion cases:                   | 20.99% (264/1258) | 19.05% (129/677)  |
> >
> > **Caption:** _Case wise error analysis of Event Split’s Train and Test sets of questions._

---

> > > ### Author Response · Authors · 2025-11-20
> > > **(4/4) A deeper analysis of the data behind Fig. 7**
> > >
> > > ### Train and Validation Sample Distributional Overlap
> > >
> > > To try and identify any additional links between the training questions and the validation questions that might explain the transfer learning indicated by the improvements in performance on validation questions, we extracted all the named entities and whitespace separated tokens (words) from the fictional documents and the answers (targets) to the questions. Then, we computed the aggregate overlap between all entities or words from different combinations of the training fictions or answers and the validation fictions or answers.
> > >
> > > The main hypothesis we pursued in this analysis is that the cases where we take the set of training questions that only became correct after training (directly “learned”) and the set of questions in the validation set where the model also happened to become correct after training (indirectly “transfer learned”) would show some elevated level of distributional overlap as compared to all possible subset comparisons. An observation like this would help explain why the model improved on those specific validation questions over others.
> > >
> > > However, we found that the entity or word based Jaccard similarity (set intersection over set union) for that specific case (row 5) no matter which combination of the fiction texts or answer texts with intersected, was never significantly any higher than the other cases in other rows. While this could simply be too simple of a distributional comparison, it does indicate that the specific improvements in validation question performance cannot be explained by particularly high levels of simple word overlap alone. Rather, there is some collection of general distributional patterns that the model is learning from the training documents that is also useful in ranking validation answers more accurately.
> > >
> > > |    |   train_init_acc |   train_final_acc |   val_init_acc |   val_final_acc |   entity_jaccard_similarity |   word_jaccard_similarity |
> > > |---:|---:|---:|---:|---:|---:|---:|
> > > |  0 |   0 |    0 | 0 |  0 |   0.062 | 0.367 |
> > > |  1 |   0 |    0 | 0 |  1 |   0.049 | 0.299 |
> > > |  2 |   0 |    0 | 1 |  0 |   0.022 | 0.183 |
> > > |  3 |   0 |    0 | 1 |  1 |   0.055 | 0.332 |
> > > |  4 |   0 |    1 | 0 |  0 |   0.066 | 0.382 |
> > > |  5 |   0 |    1 | 0 |  1 |   0.058 | 0.347 |
> > > |  6 |   0 |    1 | 1 |  0 |   0.032 | 0.234 |
> > > |  7 |   0 |    1 | 1 |  1 |   0.064 | 0.366 |
> > > |  8 |   1 |    0 | 0 |  0 |   0.053 | 0.33  |
> > > |  9 |   1 |    0 | 0 |  1 |   0.063 | 0.355 |
> > > | 10 |   1 |    0 | 1 |  0 |   0.044 | 0.301 |
> > > | 11 |   1 |    0 | 1 |  1 |   0.06  | 0.35  |
> > > | 12 |   1 |    1 | 0 |  0 |   0.064 | 0.388 |
> > > | 13 |   1 |    1 | 0 |  1 |   0.056 | 0.335 |
> > > | 14 |   1 |    1 | 1 |  0 |   0.028 | 0.216 |
> > > | 15 |   1 |    1 | 1 |  1 |   0.062 | 0.363 |
> > >
> > > **Caption**: _Fictional Documents in Train split vs. Fictional Documents in Val split_
> > >
> > > |    |   train_init_acc |   train_final_acc |   val_init_acc |   val_final_acc |   entity_jaccard_similarity |   word_jaccard_similarity |
> > > |---:|---:|---:|---:|---:|---:|---:|
> > > |  0 |   0 |    0 | 0 |  0 |   0.018 | 0.105 |
> > > |  1 |   0 |    0 | 0 |  1 |   0.014 | 0.058 |
> > > |  2 |   0 |    0 | 1 |  0 |   0     | 0.021 |
> > > |  3 |   0 |    0 | 1 |  1 |   0.016 | 0.067 |
> > > |  4 |   0 |    1 | 0 |  0 |   0.004 | 0.062 |
> > > |  5 |   0 |    1 | 0 |  1 |   0.018 | 0.071 |
> > > |  6 |   0 |    1 | 1 |  0 |   0     | 0.023 |
> > > |  7 |   0 |    1 | 1 |  1 |   0.01  | 0.046 |
> > > |  8 |   1 |    0 | 0 |  0 |   0     | 0.04  |
> > > |  9 |   1 |    0 | 0 |  1 |   0     | 0.04  |
> > > | 10 |   1 |    0 | 1 |  0 |   0     | 0.033 |
> > > | 11 |   1 |    0 | 1 |  1 |   0.009 | 0.042 |
> > > | 12 |   1 |    1 | 0 |  0 |   0.011 | 0.078 |
> > > | 13 |   1 |    1 | 0 |  1 |   0.01  | 0.059 |
> > > | 14 |   1 |    1 | 1 |  0 |   0     | 0.03  |
> > > | 15 |   1 |    1 | 1 |  1 |   0.054 | 0.082 |
> > >
> > > **Caption**: _Fictional Answers in Train split vs. Fictional Answers in Val split_
> > >
> > > |    |   train_init_acc |   train_final_acc |   val_init_acc |   val_final_acc |   entity_jaccard_similarity |   word_jaccard_similarity |
> > > |---:|---:|---:|---:|---:|---:|---:|
> > > |  0 |   0 |    0 | 0 |  0 |   0.002 | 0.022 |
> > > |  1 |   0 |    0 | 0 |  1 |   0.001 | 0.007 |
> > > |  2 |   0 |    0 | 1 |  0 |   0     | 0.003 |
> > > |  3 |   0 |    0 | 1 |  1 |   0.005 | 0.01  |
> > > |  4 |   0 |    1 | 0 |  0 |   0.003 | 0.029 |
> > > |  5 |   0 |    1 | 0 |  1 |   0.002 | 0.009 |
> > > |  6 |   0 |    1 | 1 |  0 |   0     | 0.004 |
> > > |  7 |   0 |    1 | 1 |  1 |   0.009 | 0.013 |
> > > |  8 |   1 |    0 | 0 |  0 |   0.007 | 0.044 |
> > > |  9 |   1 |    0 | 0 |  1 |   0.003 | 0.015 |
> > > | 10 |   1 |    0 | 1 |  0 |   0     | 0.006 |
> > > | 11 |   1 |    0 | 1 |  1 |   0.013 | 0.02  |
> > > | 12 |   1 |    1 | 0 |  0 |   0.003 | 0.027 |
> > > | 13 |   1 |    1 | 0 |  1 |   0.001 | 0.009 |
> > > | 14 |   1 |    1 | 1 |  0 |   0     | 0.004 |
> > > | 15 |   1 |    1 | 1 |  1 |   0.007 | 0.012 |
> > >
> > > **Caption**: _Fictional Documents in Train split vs. Fictional Answers in Val split_

---

### Meta-Review · Area_Chair_6Vrw · 2026-01-05

**Summary:**

This paper presents FictionalQA, a carefully constructed synthetic dataset for studying how language models acquire factual knowledge during training. The reviewers consistently praised the work's clarity, experimental rigor, and the importance of the research question. All reviewers acknowledged the dataset's value as a controlled testbed that enables researchers to separate factual memorization from verbatim sequence memorization through its hierarchical structure of fictional events, documents, and question-answer pairs. The primary concern raised by multiple reviewers centered on Figure 7's "leaky" generalization, where validation set performance improves alongside training set performance. The authors addressed this convincingly in their rebuttal, demonstrating through extensive analysis that this phenomenon is actually the dataset working as intended: it reveals that models learn distributional patterns rather than purely atomic facts, which is precisely the kind of insight the dataset was designed to expose. The additional analyses of entity overlap, case-wise error patterns, and sample inspection showed no special characteristics distinguishing questions the model learned in training versus validation splits, supporting the distributional learning interpretation. Minor concerns about MCQ baseline accuracy and related work contextualization were also adequately addressed through control experiments and expanded citations. Reviewer dQfK explicitly raised their score to the acceptance regime after the rebuttal, and Reviewers tcbD and 9BFP already gave strong accept scores. While Reviewer m9Sr maintained reservations, their concerns about the dataset's utility were addressed by reframing the leakiness as a feature that enables exactly the kind of mechanistic investigation the community needs. The dataset with its transparent generation pipeline, public code release, and rich annotations represents a significant contribution that will enable future research on knowledge acquisition, unlearning, and training dynamics.

**Reviewer Concerns:**

The rebuttal successfully addressed the majority of reviewer concerns through extensive additional analysis and manuscript revisions.

Fully Addressed Concerns:

The most significant concern, raised by Reviewers m9Sr, tcbD, and hkhq, was the "leaky" generalization in Figure 7 where validation performance improves without direct exposure to those events. The authors provided comprehensive analysis across four detailed comments examining case-wise error patterns, entity/word overlap statistics, and sample-level comparisons. This analysis demonstrated that the leakiness is not a flaw but rather evidence of the dataset functioning as designed to distinguish distributional learning from atomic memorization. Reviewer dQfK explicitly increased their score after this clarification.

Reviewer dQfK's concerns about related work contextualization and missing citations were addressed through manuscript revisions removing overstated claims and adding relevant references on surface-form diversity effects. The authors clarified their findings corroborate prior work in a more controlled setting.

Reviewer 9BFP's presentation concerns about underplaying contributions, dataset naming clarity, and framing relative to human cognition were all accepted and addressed with planned revisions.

Multiple reviewers questioned the above-chance MCQ baseline accuracy. The authors addressed this with control experiments showing 10-choice variants and demonstrating that improvements are strictly attributable to fictional data exposure rather than artifacts.

Partially Outstanding:

Reviewer m9Sr's fundamental concern about dataset utility remains somewhat unresolved, though their interpretation differs from other reviewers. While the extensive rebuttal analysis addressed the technical aspects, this reviewer may still view the distributional learning phenomenon as undermining the dataset's ability to isolate factual memorization. However, three other reviewers interpreted this same property as valuable for revealing learning mechanisms.

Reviewer hkhq's concern about generalization from finetuning to pretraining settings was acknowledged but not experimentally resolved, though the authors cited recent independent work using their dataset in full-scale pretraining.

**Reviewer Scores:**

Reviewer m9Sr (Original: 2 - Reject):

Despite comprehensive rebuttals addressing their technical concerns about leaky generalization, MCQ baseline accuracy, and data quality with four detailed analytical comments, this reviewer did not engage further during the discussion period. While the extensive case-wise analysis and distributional overlap studies directly responded to their primary objections, their fundamental interpretation that leakiness "undermines the dataset utility" suggests they might have remained skeptical. However, the thoroughness of the response could have shifted them toward borderline acceptance. Estimated score: 4-5 (marginally below to at threshold).

Reviewer tcbD (Original: 8 - Accept, poster):

This reviewer was already strongly positive. The authors addressed their main concern about the "troubling" leaky generalization with detailed analysis showing it reveals distributional versus atomic learning mechanisms. Other concerns about data realism and limitations placement were acknowledged and accepted for revision. Likely maintained: 8 (accept, poster).

Reviewer hkhq (Original: 6 - Marginally above threshold):

Their primary concerns about leaky generalization and MCQ evaluation were directly targeted by the extensive four-part rebuttal analysis. The authors demonstrated through entity overlap studies and case-wise error analysis that these phenomena are interpretable features rather than flaws. With no follow-up questions, the thorough response likely increased their confidence. Estimated score: 7-8 (accept).

Reviewer 9BFP (Original: 8 - Accept, poster):

Already very positive about the work's relevance and execution. Their concerns were primarily presentational—underplaying contributions, dataset naming, and human cognition framing—all of which the authors agreed to revise. These changes would strengthen but not fundamentally alter their positive assessment. Likely maintained: 8 (accept, poster).

Reviewer dQfK (Original: 4 - Marginally below threshold):

This reviewer explicitly stated in their follow-up comment: "I am increasing my score to the acceptance regime" after the authors improved contextualization with related work, calibrated claims, and provided additional empirical analysis. Confirmed new score: 6-7 (accept).

---

### Decision · Program_Chairs · 2026-01-26

Accept (Poster)